

# Quasi-Newton Methods for Atmospheric Chemistry Simulations: Implementation in UKCA UM Vn10.8

Emre Esenturk[1,2], Luke Abraham[2,3], Scott Archer-Nicholls[2], Christina Mitsakou[2,b], Paul Griffiths[2,3], Alex Archibald[2,3], John Pyle[2,3]

[1]Mathematics Institute, University of Warwick, Coventry, CV4 7AL, UK
[2]Department of Chemistry, University of Cambridge, Cambridge, CB2 1EW, UK
[3]National Centre for Atmospheric Science, Cambridge, UK
[b] Currently at the Center for Radiation, Chemical Environments and Hazards, Public Health England, Chilton, Oxon, OX11 0RQ, UK

*Correspondence to*: *Emre Esentürk*

**Abstract.** A key and expensive part of coupled atmospheric chemistry-climate model simulations is the integration of gas phase chemistry, which involves dozens of species and hundreds of reactions. These species and reactions form a highly-coupled network of Differential Equations (DEs). There exists orders of magnitude variability in the lifetimes of the different

species present in the atmosphere and so solving these DEs to obtain robust numerical solutions poses a "stiff problem". With newer models having more species and increased complexity it is now becoming increasingly important to have chemistry solving schemes that reduce time but maintain accuracy. While a sound way to handle stiff systems is by using implicit DE solvers, the computational costs for such solvers are high due to internal iterative algorithms (e.g., Newton-Raphson methods). Here we propose an approach for implicit DE solvers that improves their convergence speed and robustness with relatively

small modification in the code. We achieve this by blending the existing Newton-Raphson (NR) method with Quasi-Newton (QN) methods, whereby the QN routine is called only on selected iterations of the solver. We test our approach with numerical experiments on the UK Chemistry and Aerosol (UKCA) model, part of the UK Met Office Unified Model suite, run in both an idealized box-model environment and under realistic 3D atmospheric conditions. The box model tests reveal that the proposed method reduces the time spent in the solver routines significantly, with each QN call costing 27% of a call to the full

NR routine. A series of experiments over a range of chemical environments was conducted with the box-model to find the optimal iteration steps to call the QN routine which result in the greatest reduction in the total number of NR iterations whilst minimising the chance of causing instabilities and maintaining solver accuracy. The 3D simulations show that our moderate modification, by means of using a blended method on the chemistry solver, speeds up the chemistry routines by around 13%, resulting in a net improvement in overall run-time of the full model by approximately 3 % with negligible loss in the accuracy.

The blended QN method also improves the robustness of the solver, reducing the number of grid cells which fail to converge after 50 iterations. The differences in chemical concentrations between the control run and that using the blended QN method are negligible for longer lived species, such as ozone, and below the threshold for solver convergence almost everywhere for shorter lived species such as the hydroxyl radical.



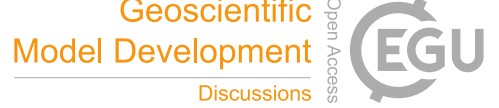

# 1 Introduction

With the advent of supercomputers, simulating the atmosphere using computational models has become an integral part of atmospheric science research, complementing experimental measurements, *in-situ* and remote observations. Model predictions

are playing an increasingly important role in both purely scientific investigations and public policy making (IPCC, 2013; Glotfelty et al., 2017). In recent years, increasing computational power has enabled the development of coupled chemistry-climate models which determine the chemical evolution and transport of trace atmospheric constituents, such as long-lived greenhouse gases, ozone, nitrogen oxides, volatile organic compounds and aerosol particles, and their influence on the environment, air quality and human health (Morgenstern et al., 2009; Heal et al., 2013; Lamarque et al. 2013; O'Conner et al.,

2014; Tilmes et al., 2015; Collins et al., 2017). These models require globally accurate predictions over time frames that span decades (Lamarque et al. 2013), involving chemical reactions of species with lifetimes ranging from sub-seconds to centuries (Whitehouse et. al., 2004), making the task computationally very expensive.

The UK Chemistry and Aerosols (UKCA) model is part of the Met Office United Model (UM) (Hewitt et al, 2011) and works as its chemistry (Morgenstern et al., 2009, O'Connor et al., 2014) and aerosol (Mann *et al.*, 2010) component. Hereafter

we refer to UM-UKCA as the fully coupled chemistry-climate model and make reference to the individual sub modules as UKCA and UM. Solving the chemistry in UKCA comes at a significant cost as it is one of the most expensive components in the UM-UKCA model. As coupled chemistry-climate models become more complex and the description of chemistry more involved, the need for computationally economic methods will be in higher demand. Hence, it makes sense to investigate ways of increasing the speed of the existing schemes with the goal of little or no sacrifice in accuracy.

Problems of a similar kind appear in other fields such as combustion systems which contain possibly reduced physical dynamics but more intensive chemistry (up to thousands of reactions) (Lu et. al., 2009) and aerosol microphysics and dynamics (Mitsakou et al, 2005). Mathematically these systems are represented by complex networks of coupled differential equations (DEs) which one must solve numerically. There is no universally best numerical method that works for every type of DE. Often one needs to choose the most reasonable method according to the need (e.g. ease of incorporating/modifying in model,

solution CPU cost/time, accuracy). The numerical methods available can be conveniently categorized as explicit or implicit. Explicit methods are quick and direct integration methods that work for many types of conventional problems but have worse stability properties, while implicit methods are more involved and indirect in calculations but have superior stability properties (Atkinson 1989). When it comes to atmospheric chemistry calculations, the main stumbling block against getting stable solutions is the problem of stiffness, which, broadly speaking, originates from different chemical reactions having orders of

magnitude different time-scales (Cariolle et. al., 2016). If one uses an explicit DE method, the (approximate) concentration values of the next timestep are calculated based on the tendencies at the current time. This makes it extremely hard to choose a timestep which is short enough to capture the chemical changes and preserve stability but also long enough to make the calculations feasible for computers. A good way to overcome this difficulty is by using an implicit method where tendencies are not based on current values, but treated as unknowns to be solved (along with the new concentration values). This greatly



increases the stability of solutions at the cost of a series of extra calculations for each timestep. But again, there is no single best implicit method which is suitable to all types of stiff problems. In fact, there are families of numerical schemes available for each category (Atkinson 1989). It is therefore desirable for any proposed new method to be flexible enough so that they can be appended to the existing solver algorithms without substantial change. This is the aim of the proposed method here.

As will be detailed further in the text, a common feature of the many currently available implicit schemes is the solution of large systems of nonlinear differential equations iteratively (Ortega and Rheinboldt, 1970; Brandt, 1977; Kelley, 1995). At each timestep, expensive subroutines have to be called several times; this is the main source of computational cost of the chemical time integration. These subroutines typically include (i) construction of a Jacobian (derivative of a function in higher dimensions) (ii) a Newton-Raphson type iterative algorithm to solve the nonlinear algebraic equations (associated to the

nonlinear differential equations). To overcome the high costs, methods that avoid or reduce Jacobian construction have gained popularity in recent years (Brown and Saad, 1990; Chan and Jackson, 1984; Knoll and Keyes, 2004). Our motivation for this work is somewhat similar in that we use approximations of the Jacobian to reduce the costs of the solver.

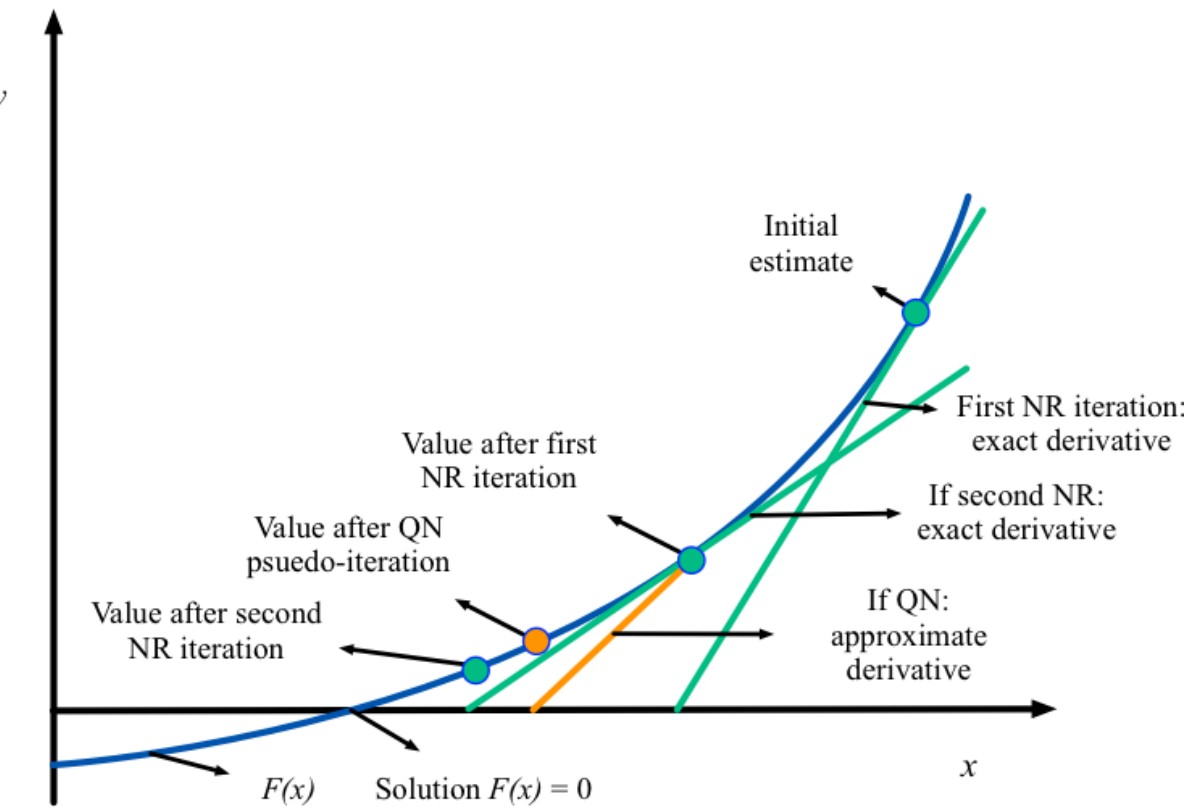

**Figure 1: Illustration of application of the QN method (adopted in our work) to find the root of a function of one variable.**



Here we develop an approach which reduces the costs of expensive routines by partly recycling the information obtained in previous iterations. We exploit them in a way that enable us to take extra steps forward for the desired solution without going through the costly parts of the cycle. The approach is an adaptation of the Quasi-Newton (QN) methods (Broyden, 1965; Shanno, 1970; Fletcher, 1970; Goldfarb, 1971; Davidon, 1991) fused into the classical Newton-Raphson (NR) method, which

are commonly used for solving large systems of nonlinear algebraic equations.

The main idea behind the QN method is illustrated in Figure 1. The objective of finding species concentrations after a short time interval can be transformed into finding the roots of a nonlinear function, which, in Figure 1, is represented as a function $F(x)$ of a single variable $x$. Numerically, the task of finding the root of the function can be achieved by the NR algorithm which is based on finding the x-intercepts following the tangent lines of values of the function (the green lines in Figure 1). The root

is obtained by simply re-evaluating the function at each x-intercept and iterating the process. The QN method uses an approximation for the tangent line (instead of an exact derivative), the orange line in Figure 1, so that computing the "new" x-intercept is quicker. In higher dimensions (e.g when solving for multiple chemical species), finding the exact derivative is equivalent to calculating the Jacobian matrix, while the QN method uses an approximate Jacobian, saving considerable computation time.

Our adaptation of the QN method uses an 'inverse update' approximation (Kvaalen, 1991) instead of the more commonly used 'forward updates' (Broyden, 1965). We demonstrate that the approach improves the convergence rate significantly and saves computational time. We further argue that using our mixed-method approach makes the algorithm more robust against "stiff environments" as it reduces the probability of the solver failing to converge on a solution and restarting using a shorter timestep. We also test how the solutions (chemical concentrations of species) are affected over a long period of integration.

We show that the differences in prognostic variables between our suggested QN method and the classical NR method are negligible and do not grow in time.

The structure of this article is as follows. In Section 2 we describe the UM-UKCA model and give a brief summary of its basic features. We then outline the current algorithm that handles the reaction kinetics by solving systems of non-linear ordinary differential equations (ODE) followed by our suggested modification using Quasi-Newton methods. We further

discuss why and how this modification works, its advantages and its possible dangers. In Section 3, we report results of our computational experiments carried out under both, a controlled box-model environment and as part of the full 3D Met Office UM-UKCA model. We compare the results of the code-modified runs with the control runs from the perspective of computational savings and differences in the concentrations/mixing ratios of chemical species, and discuss related matters with regard to parallel computing clusters. In Section 4 we conclude the paper by summarizing and highlighting our results and

pointing to possible future directions.

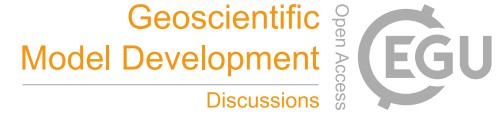

## 2 The UKCA Model

UM-UKCA, originally developed by the National Centre for Atmospheric Science, and the UK Met Office, was designed as a framework for atmospheric chemistry and aerosol computations that operates under the Met Office United Model (UM) platform and models atmospheric chemistry and aerosol fields that can feed-back onto the model dynamics via the model

radiation scheme (Morgenstern *et al.,* 2009; O'Connor *et al.,* 2014). It computes a number of possible physical-chemical processes taking place in the atmosphere such as radiation, photolysis, emissions, wet/dry deposition and clouds.  It is coupled to the UM transport dynamics sequentially, that is, transport routines and chemistry-aerosol routines are performed one after another (operator splitting) with adjustable frequency. Currently in its global configuration, for transport a timestep of 20 min is used, whilst a chemical timestep of 1hr is used to update the new concentrations of species in the model.

A number of chemical schemes are available in UKCA for modelling different parts of the atmosphere (troposphere, stratosphere etc.) with varying model details (e.g. radiative feedback switched on/off). In this paper, we use the more general stratospheric–tropospheric coupled scheme with and without an online aerosol mode (either using GLOMAP MODE (Mann et al., 2010) or aerosol climatologies) to demonstrate our results. The pure stratospheric–tropospheric mode (StratTrop) contains 75 species and consists of 283 chemical reactions (Banerjee et al., 2016). When GLOMAP MODE aerosols are

activated, 12 additional tracers are added to the system and a total of 306 reactions represent the atmospheric chemistry. The StratTrop chemical mechanism is solved using an implicit backward Euler scheme under the ASAD framework (Carver et al., 1997; Wild and Prather, 2000), as described in detail below, while photolysis is computed using the FastJ-X scheme (Wild et al., 2000). The details of these schemes can be found in Abraham et al. (2012). The UM-UKCA version used here is vn10.6.1, in the Global Atmosphere 7.1 configuration, which is a development of the UM-UKCA GA6 configuration (Walters et al,

2016).

In addition to the full 3D UM-UKCA model, we also use a box-model version of UKCA (hereafter referred as UKCA_BOX) to gain better control of the chemistry part of our simulations. UKCA_BOX is designed as a development tool using the same UKCA code, branched from version 10.1 of the UM-UKCA, but with the rest of the UM-UKCA model removed and replaced with inputs that feed the UKCA code with the same information as if it were a single grid cell in the full 3D model. The box

model uses the same StratTrop (CheST) chemical mechanism, ASAD chemical solver and FastJ-X photolysis scheme as the full 3D model, but does not have any emissions, deposition or transport. As it runs for only a single grid cell, it can be run cheaply on a single processor across many test cases. Thus, it is ideal for testing and optimising the chemical solver in UKCA over a wide range of idealised chemical environments.

In the following sections, we discuss the chemical time integration schemes in the UKCA package for determining the new

tracer concentrations and chemical tendencies. All numerical schemes are implemented using the Fortran 95 language. The code is available in the UM-UKCA trunk from version 10.8. Branches are also available at vn10.7 and vn10.6.1.

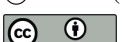



## 2.1 Chemical evolution in the UKCA

The time integration for the gas-phase chemistry in UKCA is carried out by the ASAD package which provides a flexible

framework for adding and removing new reactions/species (Carver et al., 1997; Wild and Prather, 2000). The UKCA version of the ASAD package uses a backward Euler numerical scheme to compute the new species concentrations at the next chemical time step. One of the reasons for this choice is that the relevant time scales of the reactions of species vary over many orders of magnitudes depending on the location and time of the reactions which makes the system extremely stiff. The backward Euler method is an implicit scheme which has superior numerical stability properties than almost all other explicit or semi-

explicit methods and hence works particularly well with stiff systems (Atkinson 1989). This enables the use of longer timesteps and makes long time-integrations feasible. The drawback is that, as in all implicit schemes, it demands that systems of nonlinear algebraic equations are solved at each time step, requiring extra calculations and so increasing the computational cost significantly.

These heavy costs can be partly reduced by exploiting the fact that the coupling among species is "loose" in the sense that

each species reacts with several other species but not all. This makes the Jacobian sparse and allows for the use of sparse matrix methods which significantly cuts costs. This approach was implemented in the UM-UKCA model (see Morgenstern et al, 2010).

## 2.2 Numerical implementation in the existing solver

The reaction kinetics in the atmosphere can be represented, mathematically, as a system of nonlinear ODEs where the initial values are prescribed. Emissions and dry/wet deposition enter in these equations as source and sink terms. The task of determining the change in a chemical species concentrations is equivalent to solving the coupled nonlinear system numerically.

Let $\boldsymbol{c}(t) = (c_1(t), c_2(t), \ldots, c_N(t))$ denote the vector of species concentrations at a given time. Then the species evolve

according to

$$\frac{d\boldsymbol{c}}{dt} = \boldsymbol{f}(\boldsymbol{c}) = \boldsymbol{P}(\boldsymbol{c}) - \boldsymbol{L}(\boldsymbol{c}) + \boldsymbol{E} - \boldsymbol{D}_{wet}(\boldsymbol{c}) - \boldsymbol{D}_{dry}(\boldsymbol{c}) \tag{1}$$

$$\boldsymbol{c}(0) = \boldsymbol{a}, \tag{2}$$

where **f** is the non-linear vector function (tendencies) given by the production and loss terms $\boldsymbol{P}, \boldsymbol{L}$, emissions $\boldsymbol{E}$, and wet and dry depositions $\boldsymbol{D}_{wet}, \boldsymbol{D}_{dry}$. The vector **a** is the initial concentration. The variables in bold-italic font are understood to be

vectors. To solve Equation (1) numerically using a backward Euler scheme we discretize the time variable, so the discrete equation takes the form




$$\frac{c(t_* + \Delta t) - c(t_*)}{\Delta t} = f(c(t_* + \Delta t)), \tag{3}$$

where $t_*$ is the current time and $\Delta t$ is the difference between the next chemical timestep and current time. The unknown $c(t_* + \Delta t)$ appears on both sides of the nonlinear equation which can be solved numerically using a Newton-Raphson (NR) algorithm.

## 2.3 Newton-Raphson (NR) scheme

Here we give a brief description of the NR method, which will prepare the ground for discussion of our contribution. Setting $t = t_* + \Delta t$ and $c(t_*) = c_*$ for brevity, we first write the discretized ODE (Eq. 3) in the standard form of an algebraic equation (AE), that is,

$$F(c(t)) = \frac{c(t) - c_*}{\Delta t} - f(c(t)) = 0. \tag{4}$$

The NR scheme starts with an initial guess (e.g. solution from the previous time step or a first order predictor) followed by an iteration algorithm in which the following system of linear equations is solved,

$$J(c^k)(\Delta c^k) = -F(c^k), \tag{5}$$

where $J(c^k)$ is the Jacobian at the $k^{th}$ iterate and $\Delta c^k = c^{k+1} - c^k$ is the increment (still within the same chemical timestep). At each iteration, by solving a linear equation of the form of Eq. 5, our initial guess will be improved and approaches to the actual solution of Eq. 4 as the procedure is repeated (Atkinson, 1989).

The linear equation (Eq. 5) can also be written in the form

$$(\Delta c^k) = H(c^k)F(c^k), \tag{6}$$

where $H(c^k)$ (or simply $H$) is the negative of the inverse of the Jacobian ($-J^{-1}$). This form will be particularly useful when we explain our improvement of the current method.

In the current UKCA implementation each major calculation step of the ODE solution algorithm is carried out by a separate routine as shown in Figure 2a. The main solving engine begins by calculating the current tendencies (right hand side of Eq. 1) using the updated chemical concentrations from the previous timestep (Step 1 in Figure 2a). An initial predictor guess is then calculated to be used in the following iterative loop. Then, the Jacobian is calculated using the exact quadratic form of the nonlinear reaction rates (Step 2). This step is followed by the solution of the linear Eq. 6 (Step 3). After the new increment $(\Delta c^k)$ is calculated, convergence is tested to determine whether $\Delta c^k$ is within our tolerance limit (which is set to $10^{-4}$ in the current version). If the routine passes the convergence test, the solver exits and concentrations at the next timestep are output, otherwise the process repeats until it converges on a stable solution. If the solution fails to converge after a set number of iterations (50 in the current version), is unstable, or diverges, the routine will exit and repeat using a smaller time step (typically



by halving the timestep). The expensive parts of the above procedure are, particularly, Step 2 and 3 (Figure 2a) and our goal is to reduce the number of calls to these steps as we show in the next section.

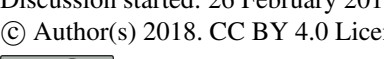

5    **Figure 2: The flowchart: flowchart showing steps taken to numerically solve the non-linear chemical equations using the Newton-Raphson method; as carried out in the standard version of ASAD in the UKCA chemical transport model (a), and in our modified version incorporating a "Quasi-Newton iteration" (b).**



## 2.4 Quasi-Newton (QN) Algorithm

We noted above that the expensive parts of the chemical integration are the Jacobian construction and solution of a system of linear equations at each iteration. Our strategy is based on the idea of using Quasi-Newton (QN) methods to minimise the number of iterations in the main Newton-Raphson (NR) solving loop, thereby reducing the number of Jacobian reconstructions and linear systems to be solved.

In QN methods, the use of exact Jacobian at every iteration is abandoned. Instead it is approximated it in a way that will satisfy certain imposed conditions. The ideas behind these (secant) methods, which date back to Broyden (1965), Shanno (1970), Fletcher (1970), Goldfarb (1971) and Davidon (1991) resemble using the inverse quotient of a function (of one variable) to replace the reciprocal of the exact derivative of the same function (see Figure 1). The price of this avoidance is a slowdown in convergence (not quadratic as in the NR algorithm, but still super-linear). In general, this strategy is more profitable since the slowdown in the convergence rate can be compensated by the substantial time gain obtained from bypassing the other costly steps compared to the time lost in the number of iterations.

Our implementation is somewhat different from the standard Quasi-Newton methods in that Newton-Raphson iterations are not completely replaced by the QN iterations. Rather, QN iterations are fused into the existing NR loop and implemented only if a chosen criterion is met. In this sense the new algorithm is a mixed method which uses both NR and QN methods as needed. This way keeps the changes to the existing algorithm minimal and makes the method flexible and practical to use. Despite this relatively small change in the algorithm the computational gain in return is considerable.

Diagrammatically (see Figure 2b), the approach works as follows: If the desired convergence has not taken place after the end of the Newton-Raphson iteration, then instead of moving on to the next iteration and reconstructing the Jacobian from scratch (Step 2), we make a pseudo-iteration and form an "effective approximation" for the inverse of the Jacobian using the concentrations already computed (Step 5). Step 6 follows in which we re-solve for the newer concentration values making use of the information available from Step 3. So, a full NR iteration is *effectively* replaced by a QN pseudo-iteration taking much less time. These measures are quantified in Section 3.1.

In the above description we refer to the "effective approximation" of the inverse of the Jacobian. However, in practice, we do not strictly construct an approximate "inverse" since taking the inverse of a matrix brings more expense. Rather, the remnants of the main NR iteration (the Jacobian from Step 2, concentrations from Step 3) are recycled and used in the approximation scheme for the inverse of the Jacobian (Broyden approximation). Schematically, after the main Newton-Raphson route we perform the following steps of 4-5-6 shown in Figure 2b which is formalized below.

We use a particular, Broyden type inverse approximation scheme (Kvaalen, 1991), which is given by the following form

$$\mathbf{H}_{app}^{k} = \mathbf{H}^{k} - \frac{\Delta c^{k} + \mathbf{H}^{k}(\Delta F(c^{k}))}{(\Delta c^{k})^{T}(\Delta c^{k})}\left(\Delta F(c^{k})\right)^{T} \tag{7,}$$





where $\Delta c^k = \mathbf{H}^k F(c^k)$ and $\Delta F(c^k) = F(c^k + \Delta c^k) - F(c^k)$ are the increments from the $k^{th}$ main iteration step, $\mathbf{H}^k$ is the inverse of the (exact) Jacobian in the main step (at the $k^{th}$ iteration) and $\mathbf{H}^k_{app}$ is the negative of the approximate inverse of the Jacobian in the pseudo-iteration after the $k^{th}$ iteration. The superscript " $...^T$ "denotes the transpose of a matrix. Although the above relation requires us to know the inverse of the Jacobian, for our purposes, we do not need to compute it explicitly.

Once $\mathbf{H}^k_{app}$ is determined, the new pseudo-increment $\delta c^k$ is given by the relation

$$\delta c^k = \mathbf{H}^k_{app} F(c^k + \Delta c^k) \tag{8}.$$

Taking $\mathbf{H}^k_{app}$ from Equation 7 and placing it into Equation 8 gives

$$\delta c^k = \mathbf{H}^k F(c^k + \Delta c^k) - (\Delta c^k + \mathbf{H}^k(\Delta F(c^k)) a^k,$$

where $a^k = \frac{(\Delta F(c^k))^T F(c^k + \Delta c^k)}{(\Delta c^k)^T (\Delta c^k)}$ . Now, recalling $\Delta F(c^k) = F(c^k + \Delta c^k) - F(c^k)$ and using the linearity of $\mathbf{H}^k$ and noting that

$\Delta c^k = \mathbf{H}^k F(c^k)$ the terms simplify

$$\delta c^k = \mathbf{H}^k F(c^k + \Delta c^k) - \mathbf{H}^k F(c^k + \Delta c^k) a^k = (1 - a^k) \mathbf{H}^k F(c^k + \Delta c^k). \tag{9}$$

If compared, we see that Eq. 9 has the same form as Eq. 6, which can also be written in the form of Eq. 5 as

$$\mathbf{J}(c^k)(\delta c^k) = (1 - a^k) F(c^k + \Delta c^k) \tag{10}.$$

Now, crucially, the information of the reduced (row echelon) matrix obtained from the original Jacobian through Gaussian

elimination in Step 3 (Eq. 5) is still available and can readily be used to solve the linear Eq. 10, where the only difference from Eq. 5 is only in the right-hand side. This bypasses the need for computing $\mathbf{H}^k_{app}$ explicitly, saving memory and time. In effect, the method accomplishes two tasks at once, reducing the combined steps of reconstructing a new Jacobian and solving a new linear equation (in a new main iteration) into a single step of solving a modified linear equation (in a pseudo-iteration) based on the information already available within that (main) iteration.

## 3. Numerical Results

In this section, we compare our results with the new method (Quasi-Newton) and without (Classical Newton-Raphson) when implemented on the current version of the UKCA solver. We consider the effectiveness of the algorithm on a single processor with, i.e., UKCA_BOX, as well as on a High Performance parallel Computing (HPC) platform (ARCHER) with the full 3D

UM simulations. In both cases our analysis will be two-fold: comparison of computational performance (savings, robustness,





etc.) and comparison of predicted model values. We show that, although the chemistry step alone takes 5% to 10% of the entire computations, there is a noticeable speed up when the chemistry component is modified in the way suggested without causing any significant error in prognostic variable values. This also improves the robustness of the computation by reducing the number of cases during the course of entire chemical integration for which the timestep has to be halved in order to converge

on a solution.

## 3.1 UKCA_BOX Simulations

To test the performance of the Quasi-Newton (QN) approximation method on performance of the UKCA chemistry solver, we first tested the changes in UKCA_BOX. UKCA_BOX allows us to test the performance of the QN methods under a highly-controlled environment, and optimise the options for the solver based on a variety of chemical conditions.

Four standard testcases were setup for these experiments to test the behaviour of the box model in different chemical environments: Urban, Rural, Marine and Stratosphere (Strat). The initial conditions for these testcases were extracted for July from a 10-year run of the full UM-UKCA model for the year 2000 at 1.875°×1.25° resolution, equivalent to the experiments conducted in section 3.2. For the Urban, Rural and Marine scenarios, average surface chemical fields, temperature, pressure and specific humidity were extracted at surface locations over the Beijing megacity, continental USA and the Pacific Ocean

respectively (see Table 1 for details). The Strat scenario used zonally averaged chemical and meteorological fields at 40°N and 32km. Full details of the scenarios are given in the supplement. The Urban scenario is initialised with the most complex mix of chemical components, and is therefore the most challenging to solve. For this reason, the analysis in the paper will focus on the Urban scenario. Results from the other scenarios are included in the supplement.

**Table 1. Summary of data points from UM model runs used to initialise UKCA_BOX scenarios, parameters describing atmospheric conditions of each scenario, and initial concentrations of select chemical species. In each case, data is extracted from a 10 year July average run of the UM-UKCA model for the year 2000.**

| Scenario | Location | Height above ground (km) | Pressure (hPa) | Temperature (K) | Specific Humidity (kg/kg) | O3 (ppbv) | NOx (ppbv) | HCHO (ppbv) |
|---|---|---|---|---|---|---|---|---|
| Urban | 40ºN, 116.4ºE | 0 | 983 | 300 | 0.0147 | 46.5 | 20.5 | 3.37 |
| Rural | 40ºN, 260ºE | 0 | 926 | 304 | 0.0101 | 49.8 | 2.5 | 1.95 |
| Marine | 40ºN, 180ºE | 0 | 1017 | 292 | 0.0121 | 25.0 | 0.30 | 0.39 |
| Stratosphere (Strat) | Zonal average at 40ºN | 32 | 8.61 | 240 | 3.44E-6 | 9,102 | 15.7 | 0.09 |





The UKCA_BOX uses the Fast-JX photolysis scheme (Wild et al., 2000), comparable to that used in the full UM-UKCA model (Telford et al., 2013). For the purposes of these experiments, a simplified setup was used whereby photolysis turns 'on' and 'off' every 12 hours of integration, using pre-calculated photolysis rates. This was done to minimise the computation of photolysis rates, and create idealised scenarios with an abrupt step-change at 'dawn' and 'dusk' to test the stability of the

solver. Photolysis rates were taken from an offline run of the 1D column Fast-JX scheme at 12 noon on $1^{st}$ July, 40°N at 0 km and 32 km in clear-sky conditions for the Urban, Rural and Marine and Strat scenarios respectively. Each experiment ran for 5 days with a 60 minute timestep (the same as the chemical timestep used in the full 3D UM-UKCA model). Without emissions, deposition or transport, the chemical evolution is completely determined by the initial conditions. Each scenario starts in a state of disequilibrium, then slowly 'winds down' over the 5 days of integration.

As discussed in the previous section, the QN method is cheaper than the full Newton-Raphson (NR) method because it does not recalculate the full Jacobian at each iteration (Table 2). On average, one QN iteration takes 27 % of the time of a full NR iteration. Since the QN method reduces the number of NR iterations required to converge, the time taken will therefore generally be reduced. However, the QN method is not as exact as the NR method, and so there is not a one-to-one efficiency: calling the QN method many times may only reduce the number of NR iterations required by a few. Finding the most efficient

setup therefore becomes an optimisation problem: how can we gain the maximum reduction in NR iterations, with as few calls to the QN method as possible? In particular, we are interested in reducing the number of iterations required for the solver during the most challenging chemical states when the equations are most stiff. This will reduce the range of time taken for cores to solve each part of the domain, therefore reducing time spent waiting for all cores to catch up to the same time in the full 3D model.

**Table 2. Wallclock times for running 1000 calls for the FN iterations and QN iteration within the UKCA box model.**

|  | Full Newton Raphson Method | Quasi-Newton Method | Ratio |
|---|---|---|---|
| CPU time for 1000 calls | 160 ± 3.1 ms | 42 ± 0.71 ms | 0.2625 |
| Wallclock time for 1000 calls | 157 ± 1.8 ms | 42.9 ± 0.15 ms | 0.273 |

To test the range of options, we devised 9 experiments for each scenario, as summarised in Table 3. The control (CNTL) experiment does not call the QN method, and is identical to the solver in the release version of UKCA. The other scenarios

call the QN method after one or more NR-iterations, as given by the numbers in the name of experiments in Table 3. For example, QN1 calls the QN Newton method after the first NR iteration only, QN2-3 calls it after the second and third NR-iterations, and QN1+ calls the QN method after every NR iteration. In general, the first iteration of the solver is where the solution is most likely to diverge and cause stability problems, and so a dampening factor of 0.5 is applied to the QN method, as is also done on the first iteration for the NR method. As shown by the flow structure of this development (Figure 2b), the

QN method is only called if the solution has not already converged.



**Table 3. Summary of experiments conducted using UKCA_BOX. The control (CNTL) experiment does not call the QN method. The other experiments call the QN method after one or more NR iterations.**

| Call QN method on | CNTL | QN1 | QN1-2 | QN1-3 | QN1+ | QN2 | QN2-3 | QN2+ | QN3 |
|---|---|---|---|---|---|---|---|---|---|
| 1st iteration: | No | Yes | Yes | Yes | Yes | No | No | No | No |
| 2nd iteration: | No | No | Yes | Yes | Yes | Yes | Yes | Yes | No |
| 3rd iteration: | No | No | No | Yes | Yes | No | Yes | Yes | Yes |
| >3rd iteration: | No | No | No | No | Yes | No | No | Yes | No |

5    Figure 3 shows chemical concentrations for a selection of chemical tracers from the box model, comparing the CNTL experiment with the QN experiments, for the Urban scenario. Similar figures for the other scenarios are included in the supplement. In this scenario, the mix of NOx and VOCs results in production of $O_3$ for the first day, then a slow loss of $O_3$ over the next four days as concentrations of short-lived tracers decay due to the lack of fresh emissions (Figure 3a). Overall, these results show the QN method is very accurate with negligible divergence from the CNTL experiment. The fractional

10    differences are largest for short-lived tracers, such as OH, but are at most of the order $10^{-5}$ or less (Figure 3f). For longer lived species, such as $O_3$ or $NO_2$, fractional changes are typically $<10^{-8}$ (Figure 3c, i). Differences between the CNTL and QN scenarios do not grow over time, rather they tend to be largest in periods which are challenging to solve (at the start of the simulation, and around dawn and dusk) and then to decay to zero.





**Figure 3. Concentrations of $O_3$, OH and $NO_2$ in molecules cm$^{-3}$ from UKCA_BOX simulations of the Urban scenario. The left panels show absolute concentrations from all scenarios, with differences too small to be observed by eye. The centre and rightmost panels show absolute and fractional differences between the CNTL and QN experiments respectively. The white band show periods with photolysis on, and grey band periods with photolysis off.**

Time series of the number of iterations required to converge for the Urban scenario are shown in Figure 4. Similar figures for the Rural, Marine and Strat scenarios are included in the supplement (Figures S1 and S2; Figures S3 and S4; Figures S56 and S6 respectively), which in general are found to converge in fewer iterations than the Urban case. The black line shows the number of NR iterations required to reach a stable solution at each timestep, and the red line shows the number of QN iterations required. The first timestep is in general the most stiff. The initial chemical concentrations are typically far from a steady state having been taken from monthly average values from model cells. The dawn and dusk periods, the timesteps immediately after





photolysis is turned on and off respectively, are the next most challenging, as changing photolysis rates causes an abrupt change in the lifetimes of many species. The inclusion of the QN method can be seen to improve the solver when the number of NR iterations (black line) is lowered compared to the CNTL scenario, and is optimal when this can be achieved with the minimum number of QN pseudo-iterations (red line, Figure 4).

The Urban scenario is the most challenging of the testcases to solve, due to the high initial concentrations of reactive tracers (Figure 4). The CNTL scenario takes 12 full NR iterations to solve the first timestep, then between 4 and 7 for each timestep thereafter, needing 4.36 iterations on average (Figure 4a). More iterations are required at dawn and dusk, with a maximum of 7 NR iterations required at dusk. Calling the QN pseudo-iteration on the first iteration (QN1, QN1-2, QN1-3 and QN1+; Figure 4b-e) reduces the number of NR iterations required to reach a stable solution on most timesteps, but increases the number of

NR iterations at dawn on most days, therefore increasing the computational costs at these timesteps compared to the CNTL run. The experiments with the QN method first called on the second iteration (QN2, QN2-3 and QN2+; Figure 4f-h) consistently reduce the number of NR iterations required to reach a stable solution. Experiment QN2-3 is the most efficient of the three, reducing the number of NR iterations required on the first timestep to 11, at the dusk timesteps to 6, and to 3.52 on average, giving a net average of 3.89 NR-equivalent iterations counting each QN psuedo-iteration as 27% of a full NR iteration.

Experiment QN2+ shows diminishing returns compared to QN2-3, calling more QN pseudo-iterations for no reduction in NR iterations on most timesteps, although it does reduce the number of NR iterations on the first timestep to 8. The experiment with QN called on the third iteration only (QN3; Figure 3i) shows only marginal improvement compared to the CNTL scenario. Overall, QN2-3 most consistently reduces the net iteration count on average and at dawn and dusk in the Urban testcase. In some of the other scenarios, QN1-3 performed most efficiently (see supplement). However, in the urban scenario the QN2-3

experiment performs better at the dawn timesteps, when the QN1-3 experiment performs worse than the CNTL run. QN1-3 therefore shows signs of reduced robustness during the periods which are most challenging to solve, meaning it is unlikely to be able to handle the wide range of chemical states that will be simulated in the 3D model runs. We therefore use the QN2-3 setup for the 3D model runs, as UKCA_BOX results suggest it shows the most consistent improvements over the CNTL scenario.





**Figure 4. Plots of solver iteration (convergence) numbers for the original full Newton-Raphson (NR) method and Quasi-Newton (QN) methods, with QN pseudo-iterations only called on particular iteration(s). The CNTL scenario (top-left) only solves with NR iterations, and is equivalent to the solver in the release version of UKCA. The other 8 panels call QN pseudo-iterations on one or more iterations at each timestep. The black lines show number of NR iterations required to converge on a stable solution, while the red line shows number of QN pseudo-iterations required. The white bands show periods with photolysis on, and grey band periods with photolysis off. The text in each panel gives the number of NR and QN iterations required to converge on the first timestep, the most difficult timestep after the first, and on average across the whole period, along with an estimate of the net computational time in NR equivalent iterations based on the assumption that one QN pseudo-iteration takes 27% of the time of a full NR iteration.**





## 3.2 UM-UKCA Simulations

In this section, we report our results for the full 3D global UM-UKCA simulations with the QN method implemented (on the original ASAD solver code) and without (classical NR method). We discuss these results from the perspectives of model performance (computational savings and stability) and prognostic evaluations (comparison of model physical values). All
simulations were performed using version 10.6.1 of the model, applying the GA7.1 configuration at 1.875°×1.25° resolution with 85 vertical levels up to 85km (*N96L85*). Emissions were year 2000 CMIP5 emissions for all runs (Lamarque et. al., 2013). Aerosols were provided via a climatology. The UM-UKCA is a non-hydrostatic model which uses a regular longitude-latitude grid and a vertical hybrid height coordinate.

  We have performed three sets of numerical experiments with two slightly different configurations of UKCA. The first version
(StratTrop) uses the stratosphere-troposphere chemistry where all radiative feedback from UKCA trace gases was turned off and aerosol climatologies were used. This set-up allows for changing the chemical species whilst maintaining the same wind fields between the simulations. The UM-UKCA is parallelised by breaking the domain up into a chess-board pattern of sub-domains, defined by the number of processes given for the East-West (EW) and North-South (NS) directions. The solver iterates across all grid cells in the sub-domain until all have reached a stable solution. Thus, the computational speed is limited
by the hardest-to-converge ("stiffest") grid cell in each subdomain. This configuration was run for 20 model years using 432-cores (24EW×18NS) in both control (CNTL) and quasi-Newton configurations (QN2-3). Additionally, four 1-year simulations were performed with additional timer diagnostics included using the Dr Hook package (ECMWF), two using 432-cores and two using 216-cores (18EW×12NS). In all these sets of simulations the initial start file was the same and the wind fields bit-compared at the end of the simulation.

A second set of simulations was performed using the stratosphere-troposphere chemistry combined with the GLOMAP-mode aerosol scheme (StratTrop+GLOMAP). This requires additional chemical species and reactions to be included on top of the standard StratTrop chemistry. In these simulations both CNTL and QN2-3 simulations were performed on 432-cores (24EW×18NS) for 20 model years (equivalent to the StratTrop simulations). However here both aerosols (via the direct and first and second indirect effects) and ozone, methane and nitrous oxide were coupled interactively to the MetUM dynamics via
the model radiation scheme. This means that the wind fields did not bit compare between these simulations as the small concentrations changes introduced by the QN method resulted in global changes to the dynamical fields. Additionally, two 1-year simulations with timer diagnostics were also completed for CNTL and QN2-3 configurations.

### 3.2.1 Model Performance

  We begin our discussion with an overview of the timing for each simulation set. These total time measurements are
complemented by a robustness assessment, checking the number of times that iteration steps of the main chemistry solver are halved in order to reach the prescribed accuracy (that is, where UKCA spends more CPU in regions of stiff chemistry). This

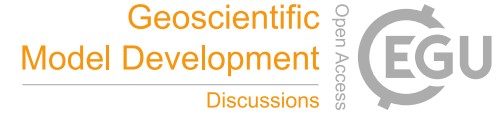


initial analysis is then expanded to a more detailed analysis via time measurement maps of the simulations and iteration maps of the chemistry solver.

Table 4 gives the total wall-clock time measurement results for the four 20-year sets of simulations (jobs). A plot of the speed up for absolute wall-clock time is also included in the supplement (Figure S7). Using our suggested modification of the

current algorithm leads to a net savings of ~2-3% over the full UM simulation despite the fact that the chemistry routine takes a relatively small part (5-10%) of the entire simulation (depending on the configuration). This suggests that using a (mixed) Quasi-Newton method has the potential to reduce the computational costs of other non-spatial systems with more intensive chemistry or even spatial systems modelled by partial differential equations that involves construction of a Jacobian for the computation of solutions. For the comparison of core components of the UKCA routines, we conducted 1-year long timer

diagnostics analysis with the Dr Hook package. The results are tabulated in Table 5. It is found that the QN scheme speeds up the chemistry component between 12.7 % and13.5 % depending on the configuration.

**Table 4. Computational speed-up using the QN method in comparison to regular Newton-Raphson method.**

| Chemistry | Number of Cores | Simulation | Mean wall-clock time for one month (s) ± 2× standard error | Speed-up (%) |
|---|---|---|---|---|
| StratTrop | 432 | CNTL | 3525.7 ± 10.6 | 2.31 ± 0.01 |
| | | QN2-3 | 3444.2 ± 9.5 | |
| StratTrop+GLOMAP | 432 | CNTL | 4805.7 ± 20.6 | 2.93 ± 0.02 |
| | | QN2-3 | 4664.7 ± 14.2 | |

**Table 5. Average relative timings of the routines in the UM-UKCA with and without the QN method (1-year jobs each).**

| Chemistry | Number of Cores | Simulation | Chemistry % | Convection % | Photolysis % | Chemistry speed-up % |
|---|---|---|---|---|---|---|
| StratTrop | 432 | CNTL | 8.65 | 3.75 | 6.21 | 13.00 |
| | | QN2-3 | 7.70 | 3.83 | 6.36 | |
| StratTrop+GLOMAP | 432 | CNTL | 6.57 | 3.29 | 4.23 | 12.47 |
| | | QN2-3 | 5.84 | 3.35 | 4.29 | |
| StratTrop | 216 | CNTL | 11.0 | 4.42 | 7.37 | 13.48 |
| | | QN2-3 | 9.89 | 4.57 | 7.64 | |



A legitimate question is to check how Quasi-Newton methods, which are essentially based on approximations, change the robustness of the numerical scheme. This is particularly important since the modelled systems are generally under stiff conditions which are prone to instability. A poorly designed approximate method could wash out important information on the direction of the chemical evolution and cause the program to crash after some number of steps. To demonstrate that the approximation scheme that we propose is safe, we show in Table 6 the number of times the UKCA model halves the timestep (a sign that the chemical conditions at that particular location and time are such that the solution to fails to converge, oscillate, or even diverge, and therefore the timestep has to be reduced). According to Table 6, with the QN modification, the occurrence of halving the timestep is nearly two times less frequent compared to the original algorithm, suggesting that the mixed QN method can be more robust in chemically stiff environments, saving more computational time overall as halving the timestep significantly increases computational costs. The parallelisation of the UM-UKCA is such that the whole model can be held up by the few grid cells which fail to converge under the normal timestep. So improving the robustness of the solver potentially has much greater benefits to net computational efficiency than just the direct reduction in cost to solve the individual grid cells.

**Table 6.** Number of times that the solver needed to halve the timestep in order to avoid divergences or wild oscillations over one year of integration.

| Chemistry | Number of Cores | Simulation | Number of halving steps | Fraction of total number of solver calls |
|---|---|---|---|---|
| StratTrop | 216 | CNTL | 457344 | 0.00288 |
| | | QN2-3 | 270101 | 0.00170 |
| | 432 | CNTL | 436048 | 0.00137 |
| | | QN2-3 | 256019 | 0.00081 |
| StratTrop+GLOMAP | 432 | CNTL | 544532 | 0.00172 |
| | | QN2-3 | 328836 | 0.00104 |

Next, we make a grid point analysis of iterations to understand the origin of computational savings. In general, the time that it takes the solver to calculate final chemical concentrations on a grid point depends heavily on the ambient photo-chemical conditions at that point and time. So, the number of iterations in which the program exits the solver loop varies significantly across the domain.

Figure 5 shows maps of the mean number of iterations to convergence (averaged over column and time) for the 1-year simulations (one chemical timestep is equal to 1 model-hour) with the StratTrop (216 and 432 cores) and GLOMAP (432 cores) schemes. The CNTL simulations (left-hand column) clearly show regions where more iterations are required. The right-





hand column shows the difference in mean number of iterations to convergence when using the QN2-3 method. Not only the mean number of iterations is reduced globally, but greater benefit is seen in the hot-spot regions noted in the CNTL simulations.

(a) StratTrop 216-core: CNTL

(b) StratTrop 216-core: QN2-3 - CNTL

CNTL: MIN=5.60 MEAN=6.59 MAX=7.60 SDEV=0.48     QN2-3: MIN=4.20 MEAN=4.76 MAX=5.39 SDEV=0.28

(c) StratTrop 432-core: CNTL

(d) StratTrop 432-core: QN2-3 - CNTL

CNTL: MIN=4.97 MEAN=6.06 MAX=7.16 SDEV=0.48     QN2-3: MIN=3.79 MEAN=4.48 MAX=5.08 SDEV=0.27

(e) StratTrop+GLOMAP 432-core: CNTL

(f) StratTrop+GLOMAP 432-core: QN2-3 - CNTL

CNTL: MIN=5.03 MEAN=6.17 MAX=7.18 SDEV=0.47     QN2-3: MIN=3.78 MEAN=4.56 MAX=5.09 SDEV=0.29

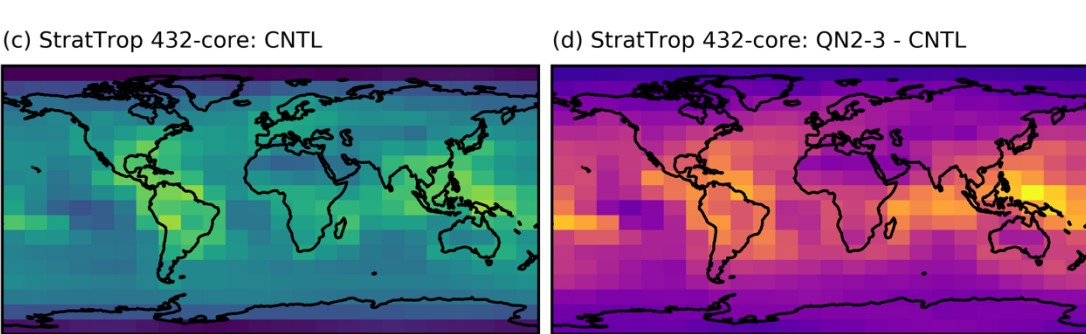

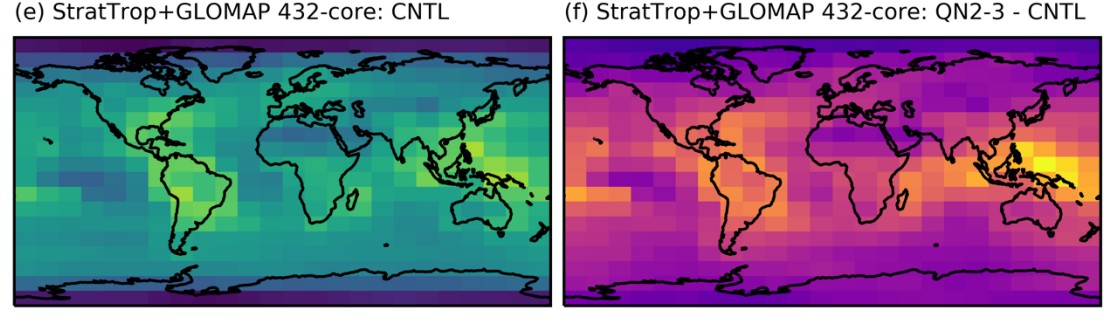

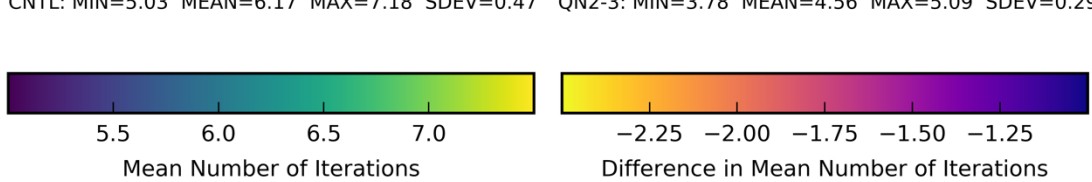





**Figure 5: : Left column (a,c,e): Maps of average iteration numbers for the 3 different 1-year standard UM Newton-Raphson solver (CNTL) simulations from Table 5, right column (b,d,f): differences between the quasi-Newton solver (QN2-3) and the equivalent control simulation. The top plots (a,b) are StratTrop 216-core (18EW×12NS), the middle plots (c,d) are StratTrop 432-core (24EW×18NS), and the bottom plots (e,f) are StratTrop+GLOMAP 432-core. The quoted statistics are for the simulations and not for the differences.**

By summing the total number of points through the 1-year year period according to number of iterations, a histogram of iteration numbers is produced which neatly summarizes performance of both methods (the CNTL and the QN cases). Figure 6 shows the histogram of the iteration numbers over all grid points for the 1-year simulations with the same StratTrop (216 and 432 cores) and GLOMAP (432 cores) schemes. The QN method greatly reduces the peak at 8 iterations, and allows the majority of solutions (approximately 70%) to be found in 4 or less iterations.

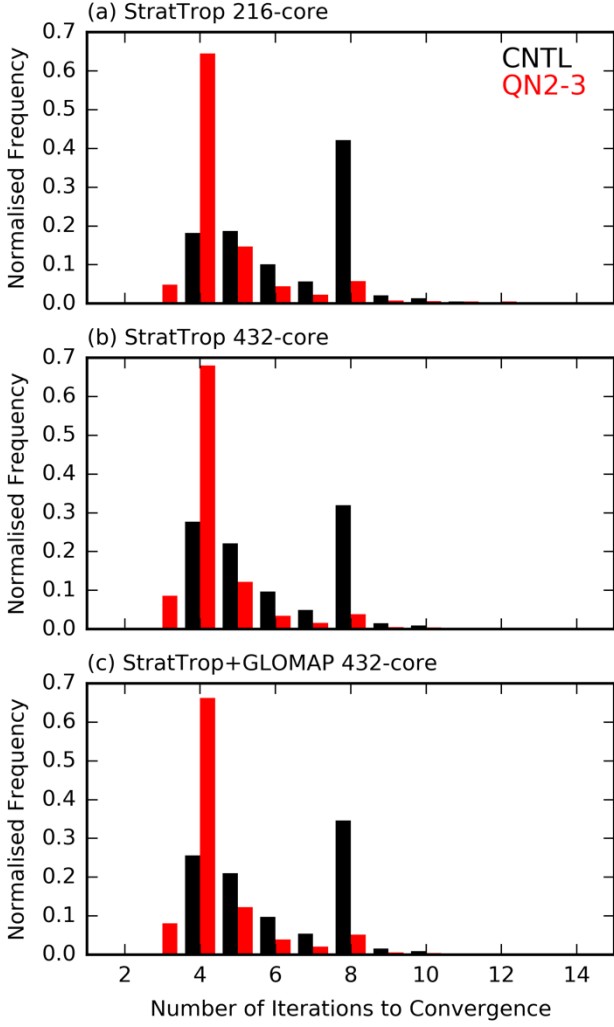

**Figure 6: Histograms of the number of iterations to convergence for the StratTrop 216-core (top), StratTrop 432-core (middle), and StratTrop+GLOMAP 432-core (bottom) 1-year long simulations.**





### 3.2.2 Model Evaluation

In this section we evaluate the accuracy of our proposed method. Recall from Section 3.1 that the QN method produces physical values which are very close to what the original method calculates even for fast changing species.

We test the accuracy of the two methods by comparing the model predictions for two different species which have very different lifetimes ($O_3$ and OH) and are key species that chemistry climate models need to simulate accurately (Monks et al., 2015). If the 3D model predictions for the two species which are on the opposite sides of the lifetime spectrum are very close, then it is very likely that physical values for all other species which have intermediate lifetimes will also be close.

For comparison of differences in values we consider only the StratTrop scenario in which ozone and other chemical
feedbacks are not included. This avoids intrinsic perturbations dominating the solutions over long periods of time and ensures that the dynamics are identical between both simulations.

From the last 10-year average of two 20-year experiments StratTrop-CNTL and StratTrop-QN2-3, we see that $O_3$ concentrations (here plotted as a 10-year mean for the representative month of July) for the two experiments are very similar, as seen in Figure 7 (for zonal-mean differences on the left column and for surface differences on the right column). The same
figures also show that the relative percentage differences (bottom row) between the two runs are negligible, being of the order of 0.01 % or smaller.

For the comparison of OH concentrations in the 20-year StratTrop-CNTL and StratTrop-QN2-3 experiments, Figure 8 shows the zonal-mean differences and surface value differences in the month of July. The difference values are slightly larger, but still only of the order of 0.1 % or smaller. Note that the largest percentage differences are seen in the areas with the smallest
absolute OH concentrations. Almost everywhere else the fractional difference in OH is less than the tolerance of the solver ($10^{-4}$). It is also clear from the surface OH plots (Figure 8 b, d, f) that the differences in OH are so small that they are approaching the limits of the numerical scheme, as the sub-domains solved by each processor are clearly visible (being 24 in the X directory and 18 in the Y direction). This artefact appears because all grid cells in each sub-domain are iterated in the solver until all have converged. and so can introduce small numerical differences.





**Figure 7: (a/c/e) Zonal-mean ozone from the last 10-years of the 20-year StratTrop 432-core simulations and (b/d/f) surface ozone from the last 10-years of the 20-year StratTrop 432-core simulations. (a/b) Ozone from the CNTL simulation. (c/d) Absolute differences QN2-3 simulation and the CNTL simulation. (e/f) Fractional differences between QN2-3 simulation and the CNTL simulation. Stippling in the (e) & (f) plots indicates that the values are below the convergence criterion of the chemical solver (10⁻⁴).**





**Figure 8: As Figure 7 but for OH. Note the use of a log-scale in the top (CNTL) plot. Note that the model domains are visible due to the extremely small differences in OH.**

## 3.2.3 Analysis of the differences between simulations with UM-UKCA

In this subsection we give a quantitative analysis of the differences in the physical values obtained from the computations. In the strict sense of the word, there is actually no extra "error" associated with our proposed method of computation as both the classical NR and QN approaches give approximate solutions of the real DE within a chosen error tolerance (which is met by each method). Nevertheless, for completeness and comparison we will regard the NR computations (CNTL runs) as the "true" values and measure the difference in OH and $O_3$ fields of the two runs using two different metrics defined below.





The figures in the previous sections provide maps of absolute and relative differences. Depending on the location of the point, these differences vary but always stay very small. In order to have a more quantitative measure of how different one particular run is from the other, we need a metric that will take into account all of the grid points and the corresponding errors. Considering the extreme low values of OH in certain regions, the most suitable metrics (Yu et. al., 2006) are the normalised

mean absolute difference (NMAD) and normalised root mean square difference (NRMSD) which are respectively defined by

$$NMAD_S = \frac{\sum_i |c_{S,nr}{}^i(T) - c_{S,qn}{}^i(T)|}{\sum_i |c_{S,nr}{}^i(T)|} \tag{11},$$

$$NRMSD_S = \sqrt{\frac{\sum_i |c_{S,nr}{}^i(T) - c_{S,qn}{}^i(T)|^2}{\sum_i |c_{S,nr}{}^i(T))|^2}} \tag{12}.$$

where $S$ denotes the species and $T$ denotes the time at the end of the run. To measure the bias we calculate normalised mean

bias which is defined as

$$NMB_S = \frac{\sum_i c_{S,nr}{}^i(T) - c_{S,qn}{}^i(T)}{\sum_i |c_{S,nr}{}^i(T)|} \tag{13}.$$

Table 7 below shows the NMAD, NRMSD and NMB for the OH and $O_3$ species.

**Table 7. Comparison of Newton-Raphson Vs Quasi-Newton methods by the metrics NMAD and NMRSD.**

| Chemistry | Species | Comparison | NMAD | NMRSD | NMB |
|---|---|---|---|---|---|
| StratTrop (432 cores) | OH | CNTL vs QN2-3 | $3.6986 \ 10^{-8}$ | $3.6019 \ 10^{-8}$ | $3.0382 \ 10^{-9}$ |
| StratTrop (432 cores) | Ozone | CNTL vs QN2-3 | $8.8374 \ 10^{-7}$ | $8.9908 \ 10^{-7}$ | $7.3761 \ 10^{-7}$ |

We also plot the NMAD, NRMSD and NMB as a function of time (each month) in the last 10-year period for OH (Figures S8, S9 and S10 of the supplement respectively) and for $O_3$ (Figures S11, S12 and S13 of the supplement respectively). We observe

that the differences are extremely small and stay bounded in time and do not grow, which indicates that the two methods reproduce essentially the same evolution. We remark that NMB values are smaller than NMAD in magnitude and do not grow in time, as expected.



## 4. Conclusion

Atmospheric chemistry simulations are at the heart of coupled chemistry-climate models. Solving the complex sets of equations that represent the evolution of species comes at a high computational cost. In this article, we introduced a version of the Quasi-Newton method into the UKCA coupled climate model. The Quasi-Newton method demonstrates improvements, in

multiple ways, over the classical Newton-Raphson method used in the UKCA model chemistry solver.

The main benefit of the QN approach, as discussed in Section 3, is its ability to reduce the computational time for the simulations. The advantages, however, are not limited to reducing the costs of chemistry calculations. The computations are more robust against stiff chemical environments thereby reducing the possibility of divergence and instability in computations. On parallel platforms, even when there is no danger of instability, robustness actually can translate into extra computational

gain as the method save further time by avoiding unnecessary wait times in the sub-domains.

We also demonstrated that, the suggested method, while improving the performance, does not deteriorate the accuracy of physical predictions, which is an obvious requirement for any proposed method. From the cross comparisons under different computational environments (UKCA_BOX or parallel UM simulations), different chemical scenarios (interactive or noninteractive) for a large spectrum of chemical species (varying from very long lifetime or short lifetime) the method

maintains the same level accuracy with the original method.

Another feature of our approach is its flexibility for use with many existing chemistry solving. Whilst this work focussed specifically on the UKCA, the algorithm can be easily integrated to the existing codes of the other (unrelated) coupled chemical system solvers. As sketched in Section 2, it is also simple to detach the algorithm from the modified program and revert back to the original algorithm if desired. Furthermore, since the method is quite generic, it can be used beyond solving chemical

systems. We think that it will be just as easy to implement the method to other components of the climate model. For instance solving systems of time dependent nonlinear (partial) differential equations which can be cast into a problem of solving systems of nonlinear algebraic equations at each timestep.

Finally, we remark that we have focused on one particular Quasi-Newton approach which took advantage of available information and use the to replace costly Jacobian construction and linear system solving routines which proved to work

robustly under fairly general conditions. There are also other Newton type methods that avoid or reduce Jacobian construction (Brown and Saad, 1990). Although these methods pursue relatively different strategies (and hence require more substantial changes to a classical NR type algorithm) it would be interesting to investigate their numerical capability.

## 4 Code Availability

Due to intellectual property right restrictions, we cannot provide either the source code or documentation papers for the UM or JULES.

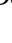


*Obtaining the UM.* The Met Office Unified Model is available for use under licence. The code is available in the UM trunk from version 10.8. Branches are also available at vn10.7 and vn10.6.1. A number of research organisations and national meteorological services use the UM in collaboration with the Met Office to undertake basic atmospheric process research, produce forecasts, develop the UM code and build and evaluate Earth system models. For further information on how to apply

for a licence see http://www.metoffice.gov.uk/research/modelling-systems/unified-model.

## Acknowledgement

We thank Oliver Wild for many useful discussions. The first author thanks Nigel Wood and Olaf Morgenstern for helpful comments. We also thank Alan Hewitt and Stuart Whitehouse for reviewing the code.

Model integrations have been performed using the ARCHER UK National Supercomputing Service and the MONSooN

system, a collaborative facility supplied under the Joint Weather and Climate Research Programme, which is a strategic partnership between the UK Met Office and the Natural Environment Research Council. This work used the NEXCS HPC facility provided by the Natural Environment Research Council. We thank NCAS for providing support for the UKCA model development.

The first and last authors were supported under the ERC (ACCI) grant (project number 267760). Alex T. Archibald and Scott

Nicholls thank the Isaac Newton Trust under whose auspices this work was funded.

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
