# Peer review of "Quasi-Newton Methods for Atmospheric Chemistry Simulations: Implementation in UKCA UM vn10.8"

_Geoscientific Model Development, 2018_

## Referee Comment (RC1) · Anonymous Referee #1 · 10 Mar 2018

This is a well crafted paper describing the numerical methods and improvements thereon with the UK global chemistry model. I recommend its reading for anyone interested in just how such models work, or in the methods used to solve small-size, non-sparse, non-linear systems. The authors apply quasi-Newtonian methods to the UK global chemistry model and carefully diagnose the consequent improvement in computational costs. It is a valuable contribution to the research community and has some important 'lessons learned'. As is the case with all such deep-model development that I have tried, the results are often not as optimistic as one hoped for when embarking, but still worth the journey. A speed-up of 13% in chemistry and 3% in total model run time is still valuable and, of course, if the other model components are worked on similarly,

the overall speedup in the chemistry becomes more important. There is one mystery in the paper that I would like explained, and some minor suggestions below.

Overall, the manuscript is written vary carefully and is easy to read. I can find hardly any typos or awkward sentences. Please fix the Table 2 title, FN should be NR I believe?

What I cannot understand is how the quasi-Newtonian method (QN), which provides only an approximation of the true Jacobian use in the standard Newton-Raphson (NR) method, is able to find an answer within the radius of convergence and thus avoid the NR method 'wandering around the wasteland of bad solutions'. I would expect this ability from Markov Chain Monte Carlo methods, but not QN. Do the authors have any explanation for this?

p.6/l.10 The deep-seated drawback of implicit schemes is that they are inherently only first-order in time. (At least all the ones I have played with). People are developing higher order tracer transport schemes (as in Lauritzen's papers), but with first-order chemical solvers, we are left with first-order errors. In fact, the authors analysis of errors in QN vs. RN later in the paper might do better to compare these with halving the time step. I think the relative errors shown here are all trivial.

p.6/l.22 (This is an aside) I am very surprised that UKCA still uses the old tricks of putting emissions and deposition into the kinetics. This is highly unstable since it forces these terms to be absorbed into a single grid cell for the full chemical time step. It becomes totally unstable with increasing vertical (or horizontal) resolution. The boundary layer is mixed on time scales of 30-90 min, and high dep velocities will empty a 50 m layer in the chemical time step. It is good that UKCA works with this coupling, but it may fail depending on intensity of local emissions or deposition.

p.10. This is a very nice derivation. In our model, we stop re-evaluating and solving the full Jacobian when our relative errors drop below some threshold (0.1 or 0.03). Within the radius of convergence, just iterating on the right-hand side vector (your c-sup-k)

[Figure]

converges almost as swiftly and much more cheaply. I had not seen the factor (1 – a-sup-k) before and will be interested to try it out. You discuss correctly about the lower operation count of QN vs NR, but it might be useful to note here the simple numerics that solving J*delta-c = -f(c) requires nˆ3 /3 operations (inverting J requires nˆ3) but that resolving this as in eqn 10 requires only nˆ2 ops.

p.12. Can you explain simply about the CPU time vs wall clock time. When we run, the CPU time is usually 6 times the wall clock time (effectively we are using 6 out of the 8 cpus on the board).

p.14/l.12. I would not have thought that 'stiffness' applied to any given iteration, but rather it was a property of the Jacobian (and the system as a whole).

p.16. Regarding the analysis of Fig 4, I would not exaggerate the 1st hour since the initial conditions (some mean values) are horribly off from the correct answer and this almost never happens in a continuously running model. The changes over twilight are a globally common problem and very important to compute efficiently.

p.18. Table 5 is a bit hard to figure out. I see that QN2-3 decreases chemistry, but its increase in convection is artificial since the total time decreased. Is there and easier way to do this? Also, it would be nice to see the timings for all the components.

p.19/l1-14. I am surprised that you allow 40 iterations with NR. Since convergence is quadratic, relative errors for the last 4 iterations go as: e-1, e-2, e-4, e-8. Thus the problem is to get within the radius of convergence (say, e-1). If our NR solver does not find a solution within 7 interations, it is time to try a new starting guess or cut the time step as you note. It would be interesting to single out and diagnose what is going wrong when a few grid cells hold up all. We found that happened when our convection lofted very-high isoprene values to the upper trop. So the fundamental problem is finding a better starting guess.

p.21. Very nice.

p.25 Table 7 is impressive, and I suspect dwarfed by the delta-t errors.

---

## Referee Comment (RC2) · Anonymous Referee #2 · 4 Apr 2018

In this paper the authors present a methodology for improving the time-to-solution for atmospheric chemistry modules. Given the large computational burden these modules carry, any robust methodology to reduce this cost is valuable. The study is certainly within scope of the journal, but I would recommend a few points are addressed prior to publication.

Abstract:

You state that 'each QN call costing 27% of a call to the full NR routine.'. I would recommend this is made more specific with regards to overall cost to the entire box-model simulation. At the moment the cost is presented as an individual function call. Whilst

an interesting statistic, and I appreciate different solvers will use different approaches to Jacobian iterations, this would be a valuable take-home message. As the authors note in other areas of the paper, some large scale models have associated costs of ∼70% total run-time from the chemical routine.

In the abstract you also state that 'The blended QN method also improves the robustness of the solver, reducing the number of grid cells which fail to converge after 50 iterations. ' Can you put this in wider context? What are the typical number of grid cells that would otherwise fail to coverge after 50 iterations? Even if the efficiency gain is small, this could be an important reason for adopting the method. I appreciate space is tight in the abstract, but do make sure this is covered elsewhere.

General points: On the general cost saving point above, do the authors feel a 3% reduction in cost is significant for the UKCA? Is this within cost variability for other implicit solvers available? This might be an interesting comparison, even within the box-model simulations. This relates to an issue with code availability, noted at the end of this review, presuming standard optimisation options can be used with standard ODE libraries.

Most Jacobian matrices in atmospheric mechanisms are sparse. Is there any perceived scaling issues with sparsity with your method?

Section points: Section 1, page 3 lines 5 onwards. Here you discuss 'construction' of a Jacobian. Defining the Jacobian of a gas phase solver is relatively easy and, whilst pedantic, Im unsure if the authors are referring to explicit calculation of a Jacobian during solver iterations or some approximation via finite difference. Please clarify since you then discuss approximating the Jacobian to reduce costs, where a finite difference approach would be more expensive than an explicit definition for any given chemical mechanism.

Section 2.3, Page 7, line 25 – is the tolerance relative and defined as a percentage?

Section 2.3. I'm interested in the strategy in the existing solver for testing convergence before reducing the time-step and re-calculating. Presumably this is a heterogeneous issue on any given domain, but how are 50 iterations chosen as a point to reduce the time-step and wouldn't an adaptive method work?

Section 3.2.1 page 18. The authors make an interesting comment on potential speedups for 'spatial systems modelled by partial differential equations'. This includes reaction-diffusion problems, or processes in particulates. However it is not clear how generally applicable the method could be. Is it possible the method could ever lead to a decrease in performance?

Section 3.2.2, page 22 line 7: 'If the 3D model predictions for the two species which are on the opposite sides of the lifetime spectrum are very close, then it is very likely that physical values for all other species which have intermediate lifetimes will also be close.' Is it not relatively easy to demonstrate the range of propagation errors in all species? I would suggest a demonstration of this is included rather than a qualitative statement on potential. If this is not straight forward, please state why.

Section 4 Line 11: 'We also demonstrated that, the suggested method..'. Please remove the comma

You also state that 'The differences in chemical concentrations between the control run and that using the blended QN method are negligible for longer lived species, such as ozone, ' Please quantify 'negligible'.

Section 4 Code Availability. Perhaps I have misunderstood licensing issues here, but this is a slightly disappointing end. Is there a reason why at least an example chemical box-model with the QN method could not be supplied? Even if this was hard-wired, without using a package such as KPP, it fits within the clear procedures and ethos now pervading all GMD papers. I'm not sure if I could reproduce your results and check the potential exciting co-benefit for other models. This could be a simple oversight, but I would suggest the authors check with the paper and editor on providing a minimum

statement on the availability of this component. Will the optional QN methods be part of a new release? Maybe I have missed this within the main body of text, or it is implicit with the paper. If this is the subject on on-going work for which the group wishes to retain IP, which is absolutely fine, then I would simply state we could all wait for some exciting follow up studies. If there is a perceived issue with general applicability, this should also be stated.

---

## Referee Comment (RC3) · Anonymous Referee #3 · 5 Apr 2018

The manuscript by Esenturk et al. describes application of a new hybrid numerical algorithm for solving the ODEs governing atmospheric chemistry simulations. The is typically a significant computational bottleneck of such simulations, although here the authors focus on chemistry-climate models whereas the issue is even more pronounced for chemical-transport models. Thus, the topic is generally of interest and suitable for GMD. The methods proposed here are not ground-breaking, but contain an incremental idea about replacing some Newton-Raphson steps with those using approximate Jacobians (i.e. quasi-Newton). The results are similarly incremental, resulting in speed ups of the overall model by 2-3%, although tests of box-model simulations show up to ~30% improvement. The authors are diligent about exploring different aspects of this

problem and considering many ways of evaluating their results, which seem to be fairly sound. I just wish… as the authors must… that the paper went further in advancing this field. The modest improvements in performance aren't likely to motivate current researches to make a switch to adapt these methods. But perhaps this is a first step in the right direction, and this type of mixed QN/NR method will have improved performance over time. I've provided detailed comments below, which essentially amount to minor revisions.

1.28: on the –> for the

1.30: can be more specific here? how many fewer grid cells?

1.31: Can be more quantitative here as well?

2.26: I object to the statement that "explicit methods are quick" which isn't necessarily true when solving a stiff system and an explicit method is forced to take times steps that are so small that the total integration cost is larger than implicit methods. But, the authors could modify there statement to be correct by changing the scope to any single internal integration step, which indeed are usually quicker for explicit methods than implicit (but still not always).

3.11: Is 2004 really a "recent year"? Or are there more recent applications to cite here?

6.15: This approach is commonly implemented in numerous models that could be cited, for example any that uses KPP-generated solvers with sparse Jacobian options enabled.

7.25: Please clarify how the increment vector is compared to the converge threshold (a scalar). Presumably some norm of the increment? State which norm and provide some justification for associated threshold value.

I can see the distinction that the authors are making between their method and typical QN, but it's a rather small difference.

12.12: Seems like replacing NR iterations with QN isn't going to always reduce the overall computational cost, as the NR step is less accurate. Does it ever (or could it ever) occur that so many more NR iterations are required that the total cost is increased?

Fig 4: Might also be useful to show the total computational expense, rather than # of iterations, as a function of timestep. Computational expense could be plotted on the right hand side axis.

15: I was a little confused as to why QN2-3 was selected based on Fig 3 and Fig 4. It seems that in terms of overall computational expense (Fig 4), there are several where the net_Avg is about 3.9 NR. I would think then these methods should be evaluated in terms of overall accuracy (Fig 3) in a comprehensive manner, using a metrics such as significant digits of accuracy ,

SDA = -log10(max_k E_k)

where E_k is the root mean square norm of the relative error compared to the CNTRL simulation, and max_k indicates that the species with maximum E_k is considered.

Then the algorithm with the fastest, most accurate, performance selected.

17.25: "did not bit" – typo?

General: The authors might consider the applicability of their work to the field of chemical transport models, which spend a much greater fraction of their wall time on solving chemistry (since transport and dynamics are not solved online).

---

## Author Comment (AC1) · 22 May 2018

We thank Reviewer 1 for his/her positive and insightful feedback. It not only helped us to improve the text but also gave us further ideas on possible ways of improving our current code for the future versions. We appreciate his/her time Our responses to specific comments are included in the supplement.
* * *

---

## Author Comment (AC2) · 22 May 2018

We thank Referee 2 for his/her valuable comments and time. Our responses to specific comments are included in the supplement.

---

## Author Comment (AC3) · 22 May 2018

We thank Referee 3 for his/her useful comments and time. Our responses to specific comments are included in the supplement.

---

## Author Comment (AC4) · 22 May 2018

We thank Referee 3 for his/her useful comments and time. Our responses to specific comments are included in the supplement.

---

## Author Comment (AC5) · 22 May 2018

We thank all reviewers for their valuable comments and time. The combined response addresses all the comments given by the reviewers. The combined response file, the revised manuscript, the revised supplement are now on the system. We are also attaching the response file to this sheet.

Please also note the supplement to this comment:
https://www.geosci-model-dev-discuss.net/gmd-2018-32/gmd-2018-32-AC5-supplement.pdf

[Figure]

**Supplement:**

**RESPONSE TO REFEREE COMMENTS for the ARTICLE**

Quasi-Newton Methods for Atmospheric Chemistry Simulations: Implementation in UKCA UM vn10.8

Emre Esenturk[1,2], Nathan Luke Abraham[2,3], Scott Archer-Nicholls[2], Christina Mitsakou[2,b], Paul Griffiths[2,3], Alex Archibald[2,3], John Pyle[2,3]

*We thank the anonymous referees for their valuable comments and time.*

**Referee 1**

*Overall, the manuscript is written very carefully and is easy to read. I can find hardly any typos or awkward sentences. Please fix the Table 2 title, FN should be NR I believe?*
*What I cannot understand is how the quasi-Newtonian method (QN), which provides only an approximation of the true Jacobian use in the standard Newton-Raphson (NR) method, is able to find an answer within the radius of convergence and thus avoid the NR method 'wandering around the wasteland of bad solutions'. I would expect this ability from Markov Chain Monte Carlo methods, but not QN. Do the authors have any explanation for this?*

The implemented the QN method which is, as the referee points out, only an approximation manages to stay in the radius of convergence partly because it is fed and backed by the NR step. One could write the code in a way that QN method alone takes the process after the initial initialisation. However, in general this will not work since, in a stiff environment like the atmosphere, after a few consecutive QN steps the approximate Jacobian will fall far from the true value and hence the program will enter the "wasteland" and the solutions will go astray. In that sense this version of the QN method utilizes the NR step instead of completely avoiding it.

We briefly mention this in the introduction and give some more information in Section 2.4 (two paragraphs before Equation 7). We added the following text to Section 1 (the second paragraph after Figure 1) in order clarify this point:

*"A key point of the implementation is that the additional internal QN iterations do not replace the NR iterations completely. Rather each QN iteration works in and is fed by the current NR iteration."*

*p.6/l.10 The deep-seated drawback of implicit schemes is that they are inherently only first-order in time. (At least all the ones I have played with). People are developing higher order tracer transport schemes (as in Lauritzen's papers), but with first-order chemical solvers, we are left with first-order errors. In fact, the authors analysis of errors in QN vs. RN later in the paper might do*

*better to compare these with halving the time step. I think the relative errors shown here are all trivial.*

Since the method is quite generic as far as the ODE scheme is concerned, it would also be possible to test and compare it with numerical schemes other than backward Euler. The current numerical scheme is partly hard wired. It can be an interesting future problem to do this comparison. We thank the referee for the new references. It is now included in the references.

*p.6/l.22 (This is an aside) I am very surprised that UKCA still uses the old tricks of putting emissions and deposition into the kinetics. This is highly unstable since it forces these terms to be absorbed into a single grid cell for the full chemical time step. It becomes totally unstable with increasing vertical (or horizontal) resolution. The boundary layer is mixed on time scales of 30-90 min, and high dep velocities will empty a 50 m layer in the chemical time step. It is good that UKCA works with this coupling, but it may fail depending on intensity of local emissions or deposition.*

While the formulation discussed here is to include emissions and deposition in with the kinetics, in fact UKCA treats emissions separately, and these are added during the boundary-layer mixing step that occurs every dynamical timestep. Dry and wet deposition are still treated as described, although dry-deposition acts throughout the whole of the boundary layer. Wet deposition follows Giannakopoulos et al 1999. We have amended the text to clarify the treatment of emissions with the following:

*"In the current implementation, emissions are treated separately during the boundary-layer mixing step, and dry deposition occurs throughout the boundary layer."*

Giannakopoulos, C., M. P. Chipperfield, K. S. Law, and J. A. Pyle (1999), Validation and intercomparison of wet and dry deposition schemes using 210Pb in a global three-dimensional off-line chemical transport model, J. Geophys. Res., 104(D19), 23761–23784, doi:10.1029/1999JD900392.

*p.10. This is a very nice derivation. In our model, we stop re-evaluating and solving the full Jacobian when our relative errors drop below some threshold (0.1 or 0.03). Within the radius of convergence, just iterating on the right-hand side vector (your c-sup-k) C2 GMDD converges almost as swiftly and much more cheaply. I had not seen the factor (1 – asup-k) before and will be interested to try it out. You discuss correctly about the lower operation count of QN vs NR, but it might be useful to note here the simple numerics that solving J\*delta-c = -f(c) requires nˆ3 /3 operations (inverting J requires nˆ3) but that resolving this as in eqn 10 requires only nˆ2 ops.*

That how much the *re-solving operation* saves, as opposed to regular solving operation, is a nice point to note. We thank the referee for pointing out/highlighting this to us. We added the following text to the end of Section 2

*"In practical terms, this means that ~N^3 numerical operations that are normally needed to solve a linear system is now reduced to ~N^2 operations which gives substantial savings within the routine. An example of the implementation of these changes is given in the pseudo-code provided in Appendix A."*

*p.12. Can you explain simply about the CPU time vs wall clock time. When we run, the CPU time is usually 6 times the wall clock time (effectively we are using 6 out of the 8 cpus on the board).*

The box-model was run with a single core, hence the wallclock was approximately equal to the CPU time. The text has been modified to say the box-model was run on a single core:

*"All UKCA_BOX experiments were run on a single processor core."*

*"Table 2. Wallclock times for running 1000 calls for the NR iterations and QN iteration within the UKCA_BOX model run on a single processor core."*

*p.14/l.12. I would not have thought that 'stiffness' applied to any given iteration, but rather it was a property of the Jacobian (and the system as a whole).*

Here we meant that the solution was more challenging to find on some timesteps than others, generally at the transition between light and dark. The text has been modified accordingly.

*"The first timestep is the most difficult to solve, as the initial chemical concentrations are typically far from a steady state having been taken from monthly average values from model cells."*

*p.16. Regarding the analysis of Fig 4, I would not exaggerate the 1st hour since the initial conditions (some mean values) are horribly off from the correct answer and this almost never happens in a continuously running model. The changes over twilight are a globally common problem and very important to compute efficiently.*

We would agree that the first timesteps are not a particularly realistic scenario, the text has been modified to focus more on the twilight steps and less on the first timestep by removing a few statements discussing analysis of the first timestep. However, we would note that the first timestep takes the greatest number of iterations to solve (12), and while rare there are many gridcells which take 12 or more to solve in the 3D model. So while the exact chemical mix of the first timestep may not be representative, the computational challenge is, and showing whether the solver can handle the first timestep or not is a good test of its robustness.

*p.18. Table 5 is a bit hard to figure out. I see that QN2-3 decreases chemistry, but its increase in convection is artificial since the total time decreased. Is there and easier way to do this? Also, it would be nice to see the timings for all the components.*

We have updated Table 5 giving the timings in seconds for the simulations performed. It is difficult to get timings for all components, as these are provided subroutine name rather than by subcomponent from the Dr Hook timing routines. We have added timings for radiation and dynamics to the table however, as well as the total time for the simulation.

The new table is as follows:

*Average wallclock time in seconds (± 2 standard error) across all processors used for various UM components comparing the CNTL and QN2-3 methods. All are from 1-year simulations performed on a Cray XC40.*

| Chemistry | StratTrop | | StratTrop+GLOMAP | | StratTrop | |
|---|---|---|---|---|---|---|
| Cores | 432 | | 432 | | 216 | |
| Simulation | CNTL | QN2-3 | CNTL | QN2-3 | CNTL | QN2-3 |
| Dynamics | 12123 ± 22 | 12099 ± 23 | 15117 ± 28 | 15297 ± 27 | 18881 ± 27 | 18743 ± 30 |
| Chemistry | 4228 ± 26 | 3678 ± 16 | 4725 ± 28 | 4123 ± 19 | 9102 ± 96 | 7875 ± 75 |
| Diagnostics | 2951 ± 1 | 2979 ± 1 | 3628 ± 1 | 3641 ± 1 | 3098 ± 1 | 3108 ± 1 |
| Photolysis | 3038 ± 7 | 3038 ± 7 | 3041 ± 7 | 3030 ± 7 | 6082 ± 43 | 6084 ± 43 |
| Convection | 1833 ± 51 | 1828 ± 51 | 2367 ± 62 | 2366 ± 62 | 3648 ± 148 | 3637 ± 148 |
| Radiation | 1184 ± 10 | 1184 ± 10 | 1140 ± 10 | 1136 ± 9 | 2487 ± 34 | 2485 ± 34 |
| UM Total | 48871 ± 0 | 47730 ± 0 | 71900 ± 0 | 70596 ± 0 | 82561 ± 0 | 79600 ± 0 |
| Chemistry Speed-up (%) | 13.00 | | 12.74 | | 13.48 | |
| UM Speed-up (%) | 2.33 | | 1.81 | | 3.59 | |

*p.19/l1-14. I am surprised that you allow 40 iterations with NR. Since convergence is quadratic, relative errors for the last 4 iterations go as: e-1, e-2, e-4, e-8. Thus the problem is to get within the radius of convergence (say, e-1). If our NR solver does not find a solution within 7 iterations, it is time to try a new starting guess or cut the time step as you note. It would be interesting to single out and diagnose what is going wrong when a few grid cells hold up all. We found that happened when our convection lofted very-high isoprene values to the upper trop. So the fundamental problem is finding a better starting guess.*

The choice of 50 iterations was made for the original formulation of the chemical solver used here, and described in Wild & Prather (2000), and has not been changed since. It should be noted that a QN step is always paired with a full NR step, and so convergence is never determined on a QN step alone.

Wild, O., and M. J. Prather (2000), Excitation of the primary tropospheric chemical mode in a global three- dimensional model, *J. Geophys. Res.*, 105(D20), 24647–24660, doi:10.1029/2000JD900399.

*p.21. Very nice.*
*p.25 Table 7 is impressive, and I suspect dwarfed by the delta-t errors.*

**We thank the reviewer for his/her positive and useful feedback. It not only helped us to improve and polish the text but also gave us further thoughts on possible ways of improving our current code for the future versions.**

**Referee 2**

*You state that 'each QN call costing 27% of a call to the full NR routine.'. I would recommend this is made more specific with regards to overall cost to the entire box-model simulation. At the moment the cost is presented as an individual function call. Whilst an interesting statistic, and I appreciate different solvers will use different approaches to Jacobian iterations, this would be a valuable take-home message. As the authors note in other areas of the paper, some large scale models have associated costs of ~70% total run-time from the chemical routine.*

The box model results show a reduction in the mean number of full NR equivalent iterations from 4.36 to 3.89, or 12% less, which resulted in a reduction in total runtime of less than 1%. The global model results show 26% reduction in the mean number of full NR equivalent iterations from 6.06 to 4.48 for 432-core StratTrop, but resulted in a 2.33% reduction in run-time. While the solver is a key part of the both the box and global models, there are many other steps such as initialisation, I/O etc., that all take time. In the 3D model, the longer simulations mean that some things are less important (initialisation for example), but the very short times for the box model make it especially sensitive to initialisation, I/O, and load on the system. The box model simulations themselves only take around 2 seconds each, and so the box model is more useful for developing a functional understanding of how the mixed-QN methods work, and as a test-bed we can use to optimise our developments, rather than as a simplified example which can be scaled to the global model, and the discussion in the paper was written to reflect this. In the box model, it can move on as soon as the chemistry in a single grid cell has converged to a stable solution, whereas in the 3D model each core needs to wait for all the other cores to finish before moving onto the next timestep, i.e the model is only as fast as its slowest member. There are important benefits from keeping processors more in-line with each other and reducing the occurrence of timestep-halving which are simply not represented in the wallclock time of the box-model. For these reasons, we consider presentation of the runtime of the box model to be misleading; the main purpose of this paper is to show how these developments improve model performance in the 3D model, not the box model.

*In the abstract you also state that 'The blended QN method also improves the robustness of the solver, reducing the number of grid cells which fail to converge after 50 iterations. ' Can you put this in wider context? What are the typical number of grid cells that would otherwise fail to converge after 50 iterations? Even if the efficiency gain is small, this could be an important reason for adopting the method. I appreciate space is tight in the abstract, but do make sure this is covered elsewhere.*

The information in Table 6 lists the number of times a halving step is performed. The QN method reduces the need for this by 40%. We have added this information to the abstract. In the configuration of UKCA used here, the solver is called over each vertical level in the domain, and we are unable to tell how many grid-cells within that layer failed to converge. The sentence in the abstract now reads:

*The blended QN method also improves the robustness of the solver, reducing the number of grid cells which fail to converge after 50 iterations by 40%.*

*On the general cost saving point above, do the authors feel a 3% reduction in cost is significant for the UKCA? Is this within cost variability for other implicit solvers available? This might be an interesting comparison, even within the box-model simulations.*

In the context of the 11,200 model years of CMIP6 simulations currently underway using the UKESM1 Earth System Model (which includes this development), a 3% saving is equivalent to saving the time to simulate over 300 model years. This is roughly 100 real days supercomputer time. Discussion of this point has been added to the conclusion:

*"Overall, we see a reduction in total computational costs of the whole UM-UKCA model of approximately 3%, corresponding to a reduction of approximately 15% in the chemistry routines. Whilst this may not seem like a big reduction, it is significant given the high costs associated with the rest of the coupled UM-UKCA model. In practice, a 3% reduction of costs for a large study involving 10,000 model years corresponds to 300 model years saved, roughly 100 real days of supercomputer-time with the current setup"*

We have not tested this method with other solvers, although it should be noted that the savings within the solver were 13%. The box model used here is designed as a tool to aid in UKCA model development, and so it is unable to run other solvers not included in UKCA.

*Most Jacobian matrices in atmospheric mechanisms are sparse. Is there any perceived scaling issues with sparsity with your method?*

We have not tested this method recently with different chemistry schemes. We have performed tests at previous UM versions using the stratospheric chemistry of Morgenstern et al (2009), the tropospheric chemistry of O'Connor et al (2014), and the StratTrop chemistry that is produced

when these two schemes are combined (as has been used in this study). These showed that the changes in run-times were essentially linear between the three schemes.

*Section points: Section 1, page 3 lines 5 onwards. Here you discuss 'construction' of a Jacobian. Defining the Jacobian of a gas phase solver is relatively easy and, whilst pedantic, I'm unsure if the authors are referring to explicit calculation of a Jacobian during solver iterations or some approximation via finite difference. Please clarify …*

For constructing the Jacobian we use explicit and exact forms *not* the finite difference approximation which would increase the costs. Where we do the approximation is in the pseudo iteration step and it is an approximation in a different sense.

*Section 2.3. I'm interested in the strategy in the existing solver for testing convergence before reducing the time-step and re-calculating. Presumably this is a heterogeneous issue on any given domain, but how are 50 iterations chosen as a point to reduce the time-step and wouldn't an adaptive method work?*

As mentioned in the response to Reviewer 1, the choice of 50 iterations was made in the original formulation of the solver prior to it being adopted by UKCA, and Table 6 shows that this halving step occurs in less than 0.3% of solver calls. Convergence is tested for any point in the vertical layer in the processor domain that has been passed to the solver, and if any gridcell within the layer is not converged the solver continues to iterate. There are a number of possible solutions to reducing the computational effort used here, and these are being actively investigated. While an adaptive method to reduce the timestep may be beneficial, it could also add more time during the checking step.

*Section 3.2.1 page 18. The authors make an interesting comment on potential speedups for 'spatial systems modelled by partial differential equations'. This includes reaction-diffusion problems, or processes in particulates. However it is not clear how generally applicable the method could be. Is it possible the method could ever lead to a decrease in performance?*

Generally the method will lead to an increase in performance unless all "future" Jacobians are constructed solely by the approximate scheme in *stiff* systems. This is the reason why, in our stiff problem, we refresh (i.e., exactly compute) the Jacobian in selected iterations. If the system is mildly stiff the method will work more effectively and one may even set $a_k=0$ in Equation 10 which amounts to using the same Jacobian for multiple times which would save even more time. However one needs to be careful as overusing the same Jacobian can cause substantial deviation from the true Jacobian and may take the iterates far off the solution.

*Section 3.2.2, page 22 line 7: 'If the 3D model predictions for the two species which are on the opposite sides of the lifetime spectrum are very close, then it is very likely that physical values for all other species which have intermediate lifetimes will also be close.' Is it not relatively easy to demonstrate the range of propagation errors in all species? I would suggest a demonstration of*

*this is included rather than a qualitative statement on potential. If this is not straight forward, please state why*

Following the suggestion of the referee we computed the normalised mean absolute difference (NMAD), normalised root mean square difference and normalised mean bias (NMB) values for all species after 10 (model) years and 20 years, and have included these in below and in the supplement and added the following sentences right before Table 7 and after Table 7 which read:

 *A complete table showing NMAD, NRMSD and NMB for all species is provided in the supplement.*

*Similar conclusions can be drawn for the other species as shown in the supplement*

**c) NMAD, NRMSD and NMB Values for all Species**

| Species | NMAD (*E-3) After 120 months | NMAD (*E-3) After 240 months | NRMSD (*E-3) After 120 months | NRMSD (*E-3) After 240 months | NMB (*E-3) After 120 months | NMB (*E-3) After 240 months |
|---|---|---|---|---|---|---|
| O3 | 0.0011 | 0.00088 | 0.0010 | 0.00090 | 0.00098 | 0.00074 |
| NO | 0.0441 | 0.0340 | 0.0066 | 0.0077 | -0.0032 | -0.0020 |
| NO3 | 0.0027 | 0.0040 | 0.0074 | 0.0170 | -0.0011 | -0.0021 |
| N2O5 | 0.0045 | 0.0027 | 0.0052 | 0.0074 | -0.0037 | -0.0016 |
| HO2NO2 | 0.0053 | 0.0059 | 0.0079 | 0.0086 | -0.0013 | -0.0015 |
| HNO3 | 0.0066 | 0.0058 | 0.0074 | 0.0116 | -0.0046 | -0.0025 |
| H2O2 | 0.0150 | 0.0172 | 0.0332 | 0.0423 | -0.0135 | -0.0158 |
| CH4 | 0.0003 | 0.0003 | 0.0004 | 0.0004 | 0.0003 | 0.0003 |
| CO | 0.0024 | 0.0025 | 0.0019 | 0.0020 | 0.0017 | 0.0017 |
| HCHO | 0.0089 | 0.0101 | 0.0276 | 0.0343 | -0.0033 | -0.0050 |
| CH3OOH | 0.0089 | 0.0105 | 0.0180 | 0.0220 | -0.0049 | -0.0064 |
| HONO | 0.0206 | 0.0204 | 0.1723 | 0.2035 | 0.0017 | 0.0040 |
| C2H6 | 0.0118 | 0.0137 | 0.0098 | 0.0129 | 0.0118 | 0.0136 |
| C2H5OOH | 0.0156 | 0.1939 | 0.0186 | 0.0279 | 0.0085 | 0.0111 |
| CH3CHO | 0.0098 | 0.0012 | 0.0010 | 0.0013 | 0.0056 | 0.0067 |

| | | | | | | |
|---|---|---|---|---|---|---|
| PAN | 0.0115 | 0.0153 | 0.0319 | 0.0530 | -0.0011 | -0.0006 |
| C3H8 | 0.0092 | 0.0127 | 0.0071 | 0.0140 | 0.0089 | 0.0123 |
| n-C3H7OOH | 0.0140 | 0.0225 | 0.0189 | 0.0412 | 0.0103 | 0.0180 |
| i-C3H7OOH | 0.0152 | 0.0237 | 0.0198 | 0.0412 | 0.0103 | 0.0180 |
| C2H5CHO | 0.0093 | 0.0134 | 0.0408 | 0.0373 | 0.0072 | 0.0101 |
| C2H6CO | 0.00933 | 0.01099 | 0.00783 | 0.00959 | 0.00926 | 0.01086 |
| CH3COCH2OOH | 0.01394 | 0.01660 | 0.01950 | 0.02661 | 0.00301 | 0.00494 |
| PPAN | 0.01001 | 0.01193 | 0.03702 | 0.06292 | -0.00931 | -0.01031 |
| CH3ONO2 | 0.01053 | 0.01355 | 0.01481 | 0.01974 | -0.00932 | -0.01286 |
| C5H8 | 0.02416 | 0.02421 | 0.02180 | 0.02464 | 0.01083 | 0.01182 |
| ISOOH | 0.02798 | 0.02933 | 0.03170 | 0.03713 | 0.01167 | 0.01110 |
| ISON | 0.03209 | 0.04159 | 0.06750 | 0.10933 | 0.00375 | 0.01014 |
| MACR | 0.02345 | 0.02557 | 0.02783 | 0.03507 | 0.01884 | 0.01998 |
| MACROOH | 0.04468 | 0.05189 | 0.05149 | 0.07872 | 0.02563 | 0.03051 |
| MPAN | 0.03073 | 0.03666 | 0.04900 | 0.06991 | -0.00697 | -0.00842 |
| HACET | 0.02630 | 0.03193 | 0.03694 | 0.05458 | -0.00538 | -0.00461 |
| MGLY | 0.14927 | 0.18045 | 0.41070 | 0.63219 | -0.14196 | -0.17186 |
| NALD | 0.04737 | 0.06276 | 0.10608 | 0.17306 | 0.02292 | 0.04034 |
| HCOOH | 0.01836 | 0.02537 | 0.02768 | 0.04264 | -0.00854 | -0.00811 |
| CH3CO3H | 0.02581 | 0.03148 | 0.03700 | 0.05843 | 0.01038 | 0.01493 |
| CH3CO2H | 0.02374 | 0.02903 | 0.03652 | 0.05757 | -0.00118 | 0.00231 |
| MVKOOH | 0.02997 | 0.03284 | 0.05591 | 0.06614 | 0.00713 | 0.00526 |
| Cl | 0.00069 | 0.00035 | 0.00090 | 0.00036 | -0.00069 | -0.00020 |
| ClO | 0.00245 | 0.00247 | 0.00738 | 0.00731 | 0.00015 | 0.00064 |
| Cl2O2 | 0.09651 | 0.05123 | 0.28837 | 0.05308 | 0.00543 | 0.02153 |
| OClO | 0.01277 | 0.01082 | 0.01654 | 0.01336 | 0.00894 | 0.00849 |
| Br | 0.00051 | 0.00056 | 0.00048 | 0.00065 | -0.00029 | -0.00023 |
| BrCl | 0.01104 | 0.00941 | 0.01503 | 0.01262 | 0.00771 | 0.00731 |
| BrONO2 | 0.00471 | 0.00439 | 0.00757 | 0.00754 | -0.00234 | -0.00186 |
| N2O | 0.00004 | 0.00003 | 0.00008 | 0.00005 | 0.00004 | 0.00003 |

| | | | | | | |
|---|---|---|---|---|---|---|
| HOCl | 0.00639 | 0.00587 | 0.01665 | 0.01657 | 0.00087 | 0.00045 |
| HBr | 0.008348 | 0.00908 | 0.01627 | 0.015961 | -0.00754 | -0.0083 |
| HOBr | 0.006061 | 0.00595 | 0.011883 | 0.012055 | 0.00258 | 0.00168 |
| ClONO2 | 0.001965 | 0.00217 | 0.00272 | 0.0031523 | -8E-05 | 0.00043 |
| CFCl3 | 4.35E-05 | 3.9E-05 | 8.4E-05 | 7.53E-05 | 4.3E-05 | 3.8E-05 |
| CF2Cl2 | 3.54E-05 | 2.7E-05 | 7E-05 | 5.03E-05 | 3.3E-05 | 2.5E-05 |
| CH3Br | 0.00044 | 0.00047 | 0.00052 | 0.00055 | 0.00044 | 0.00047 |
| N | 0.000798 | 0.00403 | 0.0006 | 0.004242 | -0.00077 | -0.0027 |
| O(3P) | 2.66E-05 | 1.3E-05 | 4.6E-05 | 3.19E-05 | -1.9E-05 | -5E-06 |
| ORGNIT | 0.056804 | 0.06181 | 0.08377 | 0.107436 | 0.03981 | 0.04092 |
| H | 8.77E-06 | 5.4E-06 | 2.7E-05 | 1.99E-05 | 8.8E-06 | 2.2E-06 |
| OH | 3.64E-05 | 3.7E-05 | 3.9E-05 | 3.6E-05 | 1.2E-05 | 3E-06 |
| HO2 | 0.001041 | 0.00108 | 0.00109 | 0.001262 | 0.0003 | 0.00023 |
| CH3OO | 0.00291 | 0.00492 | 0.00214 | 0.003857 | 0.0025 | 0.00375 |
| C2H5OO | 0.056128 | 0.0569 | 0.16649 | 0.16476 | 0.04521 | 0.04526 |
| CH3CO3 | 0.051005 | 0.05548 | 0.21106 | 0.274743 | -0.02183 | -0.0224 |
| n-C3H7OO | 0.076912 | 0.08898 | 0.2034 | 0.272403 | 0.07119 | 0.08267 |
| i-C3H7OO | 0.070862 | 0.08225 | 0.19265 | 0.258701 | 0.06177 | 0.07211 |
| C2H5CO3 | 0.032232 | 0.04066 | 0.11543 | 0.191884 | -0.01635 | -0.0173 |
| CH3COCH2OO | 0.043401 | 0.04446 | 0.11144 | 0.119976 | 0.01727 | 0.0185 |
| CH3OH | 0.005674 | 0.00593 | 0.007 | 0.009182 | -0.00167 | -0.003 |
| HCl | 0.000445 | 0.00049 | 0.00135 | 0.001333 | -8.3E-05 | -0.0001 |
| BrO | 0.001422 | 0.00155 | 0.00471 | 0.004844 | -0.00062 | -0.0006 |
| NO2 | 0.004671 | 0.00346 | 0.00718 | 0.012697 | -0.00346 | -0.002 |
| O(1D) | 2.69E-05 | 1.8E-05 | 3.2E-05 | 2.43E-05 | 1.2E-05 | 1.1E-05 |

*Section 4 Line 11: 'We also demonstrated that, the suggested method..'. Please remove the comma You also state that 'The differences in chemical concentrations between the control run and that using the blended QN method are negligible for longer lived species, such as ozone, ' Please quantify 'negligible'.*

We removed the comma and modified the text in the abstract accordingly

*blended QN method are of order $(10^7)$ for longer lived species, such as ozone, and below the threshold for solver convergence $(10^{-4})$ almost everywhere for shorter lived species such as the hydroxyl radical*

*Section 4 Code Availability. Perhaps I have misunderstood licensing issues here, but this is a slightly disappointing end. Is there a reason why at least an example chemical box-model with the QN method could not be supplied? Even if this was hard-wired, without using a package such as KPP, it fits within the clear procedures and ethos now pervading all GMD papers. I'm not sure if I could reproduce your results and check the potential exciting co-benefit for other models. This could be a simple oversight, but I would suggest the authors check with the paper and editor on providing a minimum statement on the availability of this component. Will the optional QN methods be part of a new release? Maybe I have missed this within the main body of text, or it is implicit with the paper. If this is the subject on on-going work for which the group wishes to retain IP, which is absolutely fine, then I would simply state we could all wait for some exciting follow up studies. If there is a perceived issue with general applicability, this should also be stated.*

We produced a pseudocode which is included in the appendix of the paper.

! **Pseudo Code for Solving the Equation $F(c)=0$**
! Inside the new chemistry step: determine the concentrations for the next step...

…
…
*err* $= 10^{-4}$
…

Update tendencies $(\boldsymbol{f}(\boldsymbol{c}_*))$ at the time of the current chemistry step $(t_*)$

Make an initial guess for the algebraic system as an input to the iterative solver
$\boldsymbol{c} = \boldsymbol{c}_* + \boldsymbol{f}(\boldsymbol{c}_*) \, del\_t$

! **Main NR Iteration loop starts**

! Iteration counter: k, maximum iteration counter: max_iter

*Do k=1,max_iter*

    ! Update the $\boldsymbol{F}$ vector and store it
    $\boldsymbol{F} = \frac{c - c_*}{del\_t} - \boldsymbol{f}$
    **Fold =F**

    ! Jacobian construction and linear system solving
    Compute exact Jacobian $\boldsymbol{J}(\boldsymbol{c})$ of the $\boldsymbol{F}$ vector()

    Solve for the new increment $\boldsymbol{del\_c}$ in the equation
    $\boldsymbol{J}(\boldsymbol{del\_c}) = \boldsymbol{F}$

$$err\_c = \frac{\text{maxval}(abs(\boldsymbol{delt\_c}))}{maxval(abs(\boldsymbol{c}))}$$

! Updating the **c** values
Perform treatments for troublesome convergence (e.g. $\beta$ dampening factor) or
Filtering of possible negative values in components of
$$\boldsymbol{c} = \boldsymbol{c} + \beta\, \boldsymbol{del\_c}$$

**! Test and decide if QN step will be taken**
! This can be done on iterations $2 \leq k \leq 50$, and recommended on steps 2 & 3
! This step will not be done if the **c** vector converged and the routine is about to exit

If ($err\_c \geq err$ .AND. *choice_qn*) Then

    Update the tendencies

    Update the **F** vector
$$\boldsymbol{F} = \frac{\boldsymbol{c} - \boldsymbol{c_*}}{\Delta t} - \boldsymbol{f}$$
    **delF=F-Fold**

    ! **QN approximation below** …

    Compute the Jacobian modification factor
$$a = \frac{DOT\_product(\boldsymbol{delF},\boldsymbol{F})}{DOT\_product(\boldsymbol{del\_c},\boldsymbol{del\_c})}$$

    Re-solve for the newer increment **del_c**
$$\boldsymbol{J}\,(\boldsymbol{del\_c}) = (1 - a)\boldsymbol{F}$$

    Update **c** values
$$\boldsymbol{c} = \boldsymbol{c} + \beta\, \boldsymbol{del\_c}$$

    End If

    *k=k+1*

End Do

**We thank the referee for his/her valuable comments and time.**

**Referee 3**

*The manuscript by Esenturk et al. describes application of a new hybrid numerical algorithm for solving the ODEs governing atmospheric chemistry simulations. The is typically a significant computational bottleneck of such simulations, although here the authors focus on chemistry-climate models whereas the issue is even more pronounced for chemical-transport models. Thus, the topic is generally of interest and suitable for GMD. The methods proposed here are not*

*ground-breaking, but contain an incremental idea about replacing some Newton-Raphson steps with those using approximate Jacobians (i.e. quasi-Newton). The results are similarly incremental, resulting in speed ups of the overall model by 2-3%, although tests of box-model simulations show up to ~30% improvement. The authors are diligent about exploring different aspects of this problem and considering many ways of evaluating their results, which seem to be fairly sound. I just wish. . . as the authors must. . . that the paper went further in advancing this field. The modest improvements in performance aren't likely to motivate current researches to make a switch to adapt these methods. But perhaps this is a first step in the right direction, and this type of mixed QN/NR method will have improved performance over time. I've provided detailed comments below, which essentially amount to minor revisions.*

We would like to thank anonymous Referee #3 for their time in reviewing our manuscript. We are pleased that the reviewer has found our work diligent. We set out with the ambition of increasing the efficiency of our numerical integration scheme and we are pleased with our results. For a very small increase in the number of lines of code we have achieved what we feel is a significant reduction in computational time, as seen in the UKCA_BOX modelling results. Integrating the chemical reactions remains slow but we feel that this is a forward step. Clearly we would have liked a giant leap forward but often this is not the case in science.

*1.28: on the –> for the*

*1.30: can be more specific here? how many fewer grid cells?*

As listed in Table 6, the QN method reduces this non-convergence by 40%, although the number of solver-calls where convergence is not achieved within 50 iterations is less than 0.3%.

*1.31: Can be more quantitative here as well?*

As we also replied to Referee 2, the text has been amended to:

*"The blended QN method also improves the robustness of the solver, reducing the number of grid cells which fail to converge after 50 iterations by 40%."*

*2.26: I object to the statement that "explicit methods are quick" which isn't necessarily true when solving a stiff system and an explicit method is forced to take times steps that are so small that the total integration cost is larger than implicit methods. But, the authors could modify there statement to be correct by changing the scope to any single internal integration step, which indeed are usually quicker for explicit methods than implicit (but still not always).*

We thank the referee for noting this point. We are well aware that explicit methods are not always quick. What was meant there was that explicit methods are among the handy choices to give a first go when solving non stiff systems. However, to avoid misunderstanding we modified the text to clarify the point and make it clear that explicit methods are generally quick over single time integration step. But for stiff systems they require very small time steps (to preserve stability) and

become disadvantageous. Hence, for stiff systems, despite possibly taking longer on single step, implicit methods are more advantageous for overall efficiency.

*3.11: Is 2004 really a "recent year"? Or are there more recent applications to cite here?*

We thank the reviewer for pointing this out. There are many more recent studies and we propose to add a further reference and modify the text to reflect this.

*"To overcome the high costs, methods that avoid or reduce Jacobian construction have gained popularity in recent years (Brown and Saad, 1990; Chan and Jackson, 1984; Knoll and Keyes, 2004 and citing literature e.g. Viallet et al., 2016)"*

Viallet, Maxime, Tom Goffrey, Isabelle Baraffe, Doris Folini, Chris Geroux, M. V. Popov, Jane Pratt, and Rolf Walder. "A Jacobian-free Newton-Krylov method for time-implicit multidimensional hydrodynamics-Physics-based preconditioning for sound waves and thermal diffusion." *Astronomy & Astrophysics* 586 (2016): A153.

*6.15: This approach is commonly implemented in numerous models that could be cited, for example any that uses KPP-generated solvers with sparse Jacobian options enabled.*

We thank the referee for the comment. We added the following references to KPP.

Damian, V., Sandu, A., Damian, M., Potra, F., & Carmichael, G. R. (2002). The kinetic preprocessor KPP - a software environment for solving chemical kinetics. *Computers and Chemical Engineering*, *26*, 1567–1579.

Sandu A., Verwer J. G., Blom J. G., Spee E. J., Carmichael G. R., Potra F. A., Benchmarking stiff ode solvers for atmospheric chemistry problems II: Rosenbrock solvers, Atmospheric Environment, (31), 3469-3472, 1997

*7.25: Please clarify how the increment vector is compared to the converge threshold (a scalar). Presumably some norm of the increment? State which norm and provide some justification for associated threshold value.*

Convergence is deemed to have been achieved when the maximum fractional increment over all species calculated within the layer being considered by the solver is less than $1\times10^{-4}$. This threshold has been the same for all versions of UKCA from at least UM vn6.1 onwards.

*I can see the distinction that the authors are making between their method and typical QN, but it's a rather small difference.*

It is certainly a small difference in theory, but in practice it makes a big difference when one is to choose between using purely QN steps after the initialisation or using QN selectively at the top of

NR steps. For non-stiff systems QN alone, despite being an approximation, can be used repeatedly and can be more efficient than using mixed method as here. But for our very *stiff* system, using only QN steps will almost certainly fail, or at least take longer than a pure NR approach. Therefore this distinction becomes vitally important and is a key aspect of our developments.

*12.12: Seems like replacing NR iterations with QN isn't going to always reduce the overall computational cost, as the NR step is less accurate. Does it ever (or could it ever) occur that so many more NR iterations are required that the total cost is increased?*

Yes, absolutely. This is what is meant by "diminishing returns" of increased number of QN iterations. For example, if you compare the QN2-3 scenario with QN2+ (Figure 4 g and h), at the dusk timesteps both scenarios take 6 NR iterations to solve (compared to 7 in the control run), however the QN2-3 uses a QN iterations twice (after the 2nd and 3rd NR iteration), whereas QN2+ called QN steps 4 times (after every NR iteration except the first and last). This means the QN2+ run has called the QN method two more times without reducing the number of NR iterations, such that the total cost is increased. The following sentence was extended to highlight this point:

*"However, the QN method is not as exact as the NR method, and so there is not a one-to-one efficiency: calling the QN method many times may only reduce the number of NR iterations required by a few, and in some cases calling the QN method too many times can result in a net increase in computational burden."*

This problem is why the different options for calling the QN iteration where tested, and the QN2-3 chosen as the best compromise in the widest range of situations.

*Fig 4: Might also be useful to show the total computational expense, rather than # of iterations, as a function of timestep. Computational expense could be plotted on the right hand side axis.*

Thank you, this is a good idea to more clearly highlight the take-home messages from this figure. The Figure has been edited to include a third solid black line of the net "Newton-Raphson equivalent" iterations (NR + 0.27*QN). The text discussing the figure has also been modified accordingly. This method has been used as an approximation of computational rather than actual computational expense because there is a huge amount of random variation between runs and with each call to the routine, such that the model needs to be run many thousands of times to get statistically robust average computational times (which is how the value of a QN iteration taking 27% of the tie of a NR step was derived originally).

[Figure]

**"Figure 4. Plots of solver iteration (convergence) numbers for the original full Newton-Raphson (NR) method and Quasi-Newton (QN) methods, with QN pseudo-iterations only called on particular iteration(s). The CNTL scenario (top-left) only solves with NR iterations, and is equivalent to the solver in the release version of UKCA. The other 8 panels call QN pseudo-iterations on one or more iterations at each timestep. The blue dashed lines show number of NR iterations required to converge on a stable solution, the red line shows number of QN pseudo-iterations required, and the black line total net number of NR-equivalent iterations to solved, calculated as NR + 0.27QN. The white bands show periods with photolysis on, and grey band periods with photolysis off. The text in each panel gives the number of NR and QN iterations required to converge on the first timestep, the most difficult timestep after the first, and on average across the whole period in NR-equivalent iterations."**

15: I was a little confused as to why QN2-3 was selected based on Fig 3 and Fig 4. It seems that in terms of overall computational expense (Fig 4), there are several where the net_Avg is about 3.9 NR. I would think then these methods should be evaluated in terms of overall accuracy (Fig 3) in a comprehensive manner, using a metrics such as significant digits of accuracy ,
$SDA = -log10(max_k\ E_k)$
where $E_k$ is the root mean square norm of the relative error compared to the CNTRL simulation, and $max_k$ indicates that the species with maximum $E_k$ is considered. Then the algorithm with the fastest, most accurate, performance selected.

Setting up a box-model scenario that is representative of the conditions found in the global model is actually quite challenging, and the net average is not particularly representative. The decision of which scenario to use was taken more on the basis of how many iterations were required on the 'Max' timestep than the average (i.e. how it performs in the transition between light and dark). The reason for this is because in the global model, it can only proceed once all cores have completed their chemistry (see response to Reviewer #2 comment #1). As computing chemistry at dawn/dusk are typically the most challenging periods to solve, and as there will always be processors along the terminator somewhere in the world, the time taken to solve the "Max" step was considered more representative of the runtime of the global model than the simple average. Scenario QN2-3 has the lowest "Max" value in Figure 4 compared to all the other scenarios. The following discussion has been added to highlight this point:

*"While the UKCA_BOX model only solves a single case at any one timestep, each core in the 3D model will solve for many gridcells at each timestep, and can only move on to the next timestep once all have converged. In other words, the 3D model is only as fast as its slowest gridcell. For this reason, the cases where the new methods reduce iteration count at the more challenging timesteps (at dawn and dusk) are considered a stronger indication that they will improve integration time in the full 3D model than the average."*

Regarding the accuracy, the error of all of the scenarios were considered small enough to not be of consequence in making the decision. Errors in OH were some of the highest, and still typically $< 10^{-5}$ (SDA ~ 5) for all scenarios (Figure 3f). The largest SDA for all species was for OClO in the QN1+ scenario, with RMSE = $3.47^{-05}$ and SDA = 4.5. Given how this development has been coded, one would not expect the maximum error in any method to be more than $10^{-4}$ (SDA=4), because the same test is used to exit the solver in every case (when the maximum increment over all species is less than $10^{-4}$ after a Newton-Raphson iteration). The tolerance of $10^{-4}$ is the same as that used by default in the global model. We would be concerned if any of the experiments showed an error $>10^{-4}$ and/or diverged over time relative to the control, and none of the scenarios show this.

This should read "did not bit-compare". To clarify this statement we have amended to text to:

*"This means that the wind fields in these simulations were not identical as the small concentration changes introduced by the QN method resulted in global changes to the dynamical fields."*

*General: The authors might consider the applicability of their work to the field of chemical transport models, which spend a much greater fraction of their wall time on solving chemistry (since transport and dynamics are not solved online).*

We thank the reviewer for this point. Indeed, we feel that the our hybrid approach would provide a bigger saving for models that are not calculating full atmospheric physics calculations and as

we have shown with the UKCA_BOX results, significant speed ups can be realised. This point has been added to the conclusions:

*"If implemented in a chemical transport model, for example, one would expect the overall benefit to be greater, due to the greater proportion of computational expense of the chemical solver due to the lack of other online physical processes."*

**We thank the referee for his/her useful comments.**
* * *
In addition to changes suggested by the referees we also made few small changes such as fixing typos and small textual changes to improve clarity which are all indicated/highlighted in the revised document (line numberings are according to the original document).

*IN THE MANUSCRIPT*
- *Pg 1 ln 2, Replaced "Vn" with "vn" in the heading*
- *Pg 1, the authors' affiliations have been clarified*
- *Pg1 ln 3, added the first name Nathan for Luke Abraham*
- *Pg 2 ln9, moved the Morgenstern reference slightly above*
- *Pg 4 ln 2, reworded the sentence as "...* generated within the iterations. The method is based on exploiting this information in a way that enable one*…"*
- *Pg 4 ln 16, added the text "with respect to number of main NR iterations"*
- *Removed commas on pg 5, ln 15 and 23. Pg 28 ln 15.*
- *Pg 6 removed the whole blank line before Section 2.1*
- *Deleted 'a' pg 6 ln 23*
- *Pg 7 before Section 2.3, added the text "the vector of species concentrations at the next chemical time step"*
- *Pg 7 ln ln 13, added the text "(or simply J^k)"*
- *Pg 7 ln18, replaced H with H^k and replaced the text (–J)^(-1) with (-J^k))^-1*
- *Pg 7 ln 22 Clarified type of initial predictor (forward Euler).*
- *pg 7 ln 25, Clarification the convergence is tested by a "relative change":*
- *Added commas: pg 10 ln15, ln 24. Pg 28 ln 27*
- *Pg 21, 22 and 23, Figures 5 and 6: clarified "iteration numbers" refers only to number of Newton-Raphson iterations (NR iterations).*
- *Moved "is" in sentence, pg 20, ln 2*
- *Pg 26 ln 19, (conclusion) we added the text "using options defined in the namelist"*

*IN THE SUPPLEMENT*
- *Pg 1, changed BoxModel->BOX_MODEL, FN iterations-> NR iterations*
- *Figures S2, S4, S6 are updated in the same way as in the main text*
- *Pg 11, caption of Figure S8, "OH" is replaced with "Ozone"*
- *Pg 12, added a new subsection containing the NMAD, NRSMD, NMB values of all species*